

**Identifying MBL cloud boundaries and phase over the Southern Ocean using airborne**
**radar and in-situ measurements during the SOCRATES campaign**
Anik Das[1], Baike Xi[1], Xiaojian Zheng[1,a], and Xiquan Dong[1]*
[1]Department of Hydrology and Atmospheric Sciences, University of Arizona, Tucson, AZ,
USA
[a]Now at Environmental Science Division, Argonne National Laboratory, Lemont, IL, USA
**Correspondence:** Xiquan Dong (xdong@arizona.edu)
**Abstract.**
The Southern Ocean Clouds, Radiation, Aerosol Transport Experimental Study (SOCRATES)
was an aircraft-based campaign (Jan 15 – Feb 26, 2018) using in-situ probes and remote
sensors, targeting low-level clouds over the Southern Ocean (SO). A novel methodology was
developed to identify cloud boundaries and classify cloud phases in marine boundary layer
(MBL) clouds using airborne HIAPER Cloud Radar (HCR) and in-situ CDP+2D-S
measurements. Cloud boundaries were determined using HCR reflectivity and spectrum width
gradients. Single-layer low-level clouds accounted for ~85% of observed cases. HCR-derived
boundaries showed decent agreement with the Ceilometer and Micropulse lidar (MPL)-
measurements during the Measurement of Aerosols, Radiation, and Clouds (MARCUS) ship-
based campaign, with mean base and top differences of 0.04 km and 0.29 km. Additionally,
HCR-derived cloud base heights correlated well (R = 0.78) with HSRL observations. A
reflectivity–liquid water content (Z-LWC) relationship, $LWC = 0.70Z^{0.29}$, was derived to
retrieve LWC and liquid water path (LWP) from HCR profiles. The estimated LWP closely
matched MARCUS microwave radiometer (MWR) retrievals, with a mean difference of 9.24
g/m². Cloud phase was classified using HCR-measurements, temperature, and LWP. Among
single-layered LOW clouds, 48.8% were classified as liquid, 23.3% mixed-phase, and 6.9%
ice, with additional categories identified: drizzle (16.2%), rain (3.4%), and snow (1.5%). The
classification algorithm demonstrated over 90% agreement with established phase detection
methods. This study provides a robust framework for boundary and phase detection of MBL
clouds, offering valuable insights into cloud microphysical processes over the SO and
supporting future efforts in satellite algorithm development and climate model evaluation.
**1. Introduction**

The Southern Ocean (SO) clouds contain mainly low-level clouds, which significantly
influence the regional radiation budget (60°S latitude, encircling Antarctica). Yet global
climate models struggle to simulate them accurately (Bodas-Salcedo et al., 2016; Kay et al.,
2016; Trenberth & Fasullo, 2010; Wang et al., 2018, McCoy et al., 2014; D'Alessandro et al.,
2021). Low-level marine boundary layer (MBL) clouds over the SO exhibit a high prevalence
of SLW, with ~80% containing SLW across a temperature range of -40 to 0°C (Hu et al., 2010).
Their macrophysical and microphysical properties differ significantly from subtropical MBL
clouds, with dominant warm liquid clouds (Dong et al., 2014; Wu et al, 2020; Zhao et al.,
2020), and from Arctic mixed-phase clouds, which feature liquid tops and ice-dominated bases
(Qiu et al., 2015; Jackson et al., 2012; Moser et al., 2023). Understanding the dominant cloud
phase and spatial homogeneity of low-level SO clouds is critical for improving cloud



parameterizations and refining global climate model predictions (Zhao et al., 2023; Liu et al., 2023).

Identifying cloud phase is crucial for accurately retrieving cloud macrophysical and microphysical properties, as most retrieval algorithms are phase- and region-specific (Shupe, 2007). Various methods have been developed for classifying cloud type, phase, and hydrometeors over the SO (e.g., Xi et al., 2022; Desai et al., 2023; Schima et al., 2022; D'Alessandro et al., 2021, 2019; Romatschke & Vivekanandan, 2022; Atlas et al., 2021; Zaremba et al., 2020) and Arctic clouds (e.g., Shupe, 2007; Korelov & Milbrandt, 2022; Mioche et al., 2015; Matus & L'Ecuyer, 2017), each with varying performance based on retrieval methods and assumptions. Compared to ground-based measurements, aircraft in-situ observations provide more reliable datasets, minimizing retrieval uncertainties by directly sampling cloud boundaries and interiors. Additionally, onboard radar and lidar experience less attenuation than ground-based sensors (Ewald et al., 2021). Cloud phase retrieval remains highly dependent on observational scale, sample size, and viewing direction. Ground-based radar and lidar more accurately capture low-level clouds but suffer attenuation when observing higher altitudes. On the other hand, satellite-based sensors effectively detect high-level clouds but may struggle with low-level cloud retrieval due to signal attenuation (Dong et al., 2025).

The Southern Ocean Clouds, Radiation, Aerosol Transport Experimental Study (SOCRATES) provided a valuable dataset for investigating marine boundary layer (MBL) clouds over the SO. The airborne in-situ probes— Cloud Droplet Probe (CDP) and Two-Dimensional, Stereo, Particle Imaging Probe (2D-S), and remote sensors— the 94.4 GHz (W-band) HIAPER Cloud Radar (GV-HCR) and the 532 nm High Spectra Resolution Lidar (NCAR-HSRL) onboard the research aircrafts during SOCRATES enabled direct observations of precipitation, cloud particles, and aerosols across different cloud layers, providing vertical profiles to characterize MBL structure and the free troposphere. Previous studies utilizing SOCRATES measurements for cloud phase classification include a multinomial logistic regression (MLR) method by D'Alessandro (2021) which used in-situ cloud and drizzle probe (CDP & 2D-S) measurements to estimate cloud phase heterogeneity and frequency distributions. This method identified significant SLW and ice-phase clouds and was later refined by Schima (2022) to address inconsistencies. Romatschke & Vivekanandan (2022) presented a fuzzy logic scheme that classifies cloud hydrometeor types as a time-height profile using cloud radar-lidar data. Atlas (2021) developed the University of Washington Ice-Liquid Discriminator (UWILD), a random forest-based single-particle phase classification method for binary 2D-S images.

Remote sensing offers 2-dimensional cloud profiles, complementing the size-resolved distributions captured by in-situ measurements. However, relying solely on either method can introduce discrepancies—in-situ probes detect particles only at the aircraft's altitude, while radar-lidar profiles provide vertical cloud structure but suffers near-surface contamination due to surface clutter (Dong et al., 2025). Therefore, integrating in-situ sampling with remote sensing provides significant advantages for studying atmospheric processes (Wang et al., 2012). Lidar, with its shorter wavelength, resolves aerosols, ice precipitation, optically thin clouds, and cloud boundaries (Wang et al., 2012, 2009; McGill et al., 2002), but is easily attenuated by thicker cloud layers, such as liquid clouds (Sassen, 1991). Therefore, this study focuses on developing a method solely based on radar measurements to identify cloud boundaries and phase in MBL clouds.

In this study, we aim to use a combination of both in-situ and radar-based measurements during SOCRATES to:

1. Develop a method to identify cloud base and top heights using airborne HCR radar reflectivity and spectrum width gradient, offering cloud boundary estimation without radiosonde or drop-sonde data. Further, derive an LWC-Z exponential relationship



from in-situ measured liquid water content (LWC) and calculated reflectivity (Z) from
CDP and 2D-S probes and apply it to HCR reflectivity profiles to obtain radar-based
LWC and liquid water path (LWP).
2. Develop a cloud phase estimation method for classified low-level clouds sampled
during SOCRATES, using a combination of HCR measurements, temperature profiles
and estimated LWPs. Compare the classified phase categories with existing products
over the SO.
The manuscript is organized in the following manner: Section 2 contains a brief
introduction to the SOCRATES campaign, descriptions of the remote sensors and in-situ
probes, along with a LWC-Z relationship derived using in-situ measured LWC and calculated
reflectivity (Z) from in-situ measurements. The classification methodology for low-level
clouds based on cloud-base and -top heights is described in Section 3. Section 4 contains the
cloud phase classification methodology along with an extensive evaluation through
comparisons with existing phase-determination methods and results from other campaigns over
the SO. Finally, summary and conclusions are presented in Section 5.

## 109  2 SOCRATES observations and deriving LWC-Z relationship

### 110  2.1 SOCRATES in-situ and remote sensing datasets

A brief overview of the SOCRATES aircraft field campaign is provided in Section S1, with
a list of in-situ probes and radar-lidar instruments onboard the research aircraft in Table S1 of
the Supplementary. This study primarily utilized the measurements from two airborne in-situ
probes—CDP (Lance et al., 2010; Faber et al., 2018) and 2D-S (Lawson et al., 2006, Wu and
McFarquhar, 2019), along remote sensors— GV-HCR (NCAR/EOL HCR Team., 2014,
Romatschke et al., 2021, Vivekanandan et al., 2015) and NCAR-HSRL (NCAR/EOL HSRL
Team., 2012, Eloranta., 2005, Su et al., 2008), aboard the Gulfstream-V (GV) research aircraft
during SOCRATES. The bulk cloud microphysical properties (LWC, particle size
distributions, and number concentration) were derived from the CDP and 2D-S measurements,
which were merged into a continuous dataset with size bins from 2 to 40 µm for cloud droplets
and 40 to 1280 µm for drizzle particles, at 1 Hz temporal resolution for each flight. Following
Zheng (2024), the CDP and 2D-S datasets were combined into a single size distribution, with
droplet number concentrations in the overlapping size bin redistributed using a gamma
distribution, ensuring a complete cloud and drizzle size spectrum.
The HCR reflectivity (dBZ), Doppler velocity ($V_d$) (m/s), and spectrum width (WID)
(m/s) along with the HSRL measured backscatter coefficient ($\beta$) ($m^{-1}sr^{-1}$) and particle
depolarization ratio (PLDR) were collected at 1 Hz temporal resolution. The HSRL signals are
highly sensitive to cloud droplet concentrations and can be attenuated within a few hundred
meters in liquid cloud layers (Ewald et al., 2021; Sassen, 1991). Estimated instantaneous
uncertainties for HSRL measurements at 532 nm are 0.36 for backscatter ($\beta$) and 0.009 for
depolarization ($\delta$) (Su et al., 2008). Due to signal attenuation, HSRL detects fewer clouds than
HCR, particularly in optically thick clouds. The radar-lidar overlap is about 11% when
considering full time-height (3D) cloud profiles. Given this limitation, lidar signals are not used
for phase or boundary estimation in optically thicker MBL clouds. Cloud temperatures were
provided by the 2-dimensional ERA5 reanalysis product which matched the vertical and
temporal resolution of the HCR data (NCAR/EOL HCR Team, 2023). The HCR dataset was
further filtered to retain only nadir or zenith pointing observations, excluding all cloud samples
in transition or rotational pointing directions (i.e., those not equal to ±90 degrees).




## 2.2 Estimating LWC-Z relationship and LWP from in-situ measurements


The cloud-droplet number concentration and particle size distribution from the merged
CDP+2D-S dataset is used to calculate in-situ reflectivity factor (Z, mm$^6$/m$^3$) and liquid water
content (LWC, g/m$^3$) for cloud and drizzle particles, using the equations (Doviak et al., 1993;
Kang et al., 2021; Zheng et al., 2024) as follows:

$$Z = \int_{D_{min}}^{\infty} N(D)D^6 dD \qquad (1)$$

$$LWC = \rho_w \frac{\pi}{6} \int_{D_{min}}^{\infty} N(D)D^3 dD \qquad (2)$$

where $\rho_w$ is the density of liquid water, $D$ is the particle diameter measured as droplet size
distribution (DSD) from the CDP+2D-S particle size bins, and N(D) is the number
concentration (#/cm$^3$) per bin. Z (mm$^6$/m$^3$) can further be converted to dBZ as *dBZ = 10log(Z)*.
Clouds is defined when LWC is greater than 0.01 g/m$^3$ and cloud droplet number
concentration ($N_c$) is greater than 5 cm$^{-3}$ (Zheng et al., 2024). The LWC threshold ensures
sufficient cloud density and number concentration while removing clear-sky conditions and
aerosol noise. The number concentration of ice particles > 200 μm is very low (Zheng et al.,
2024), suggesting most ice-phase particles fall below the 2D-S-defined 200 μm threshold for
ice particle size distribution (Wu & McFarquhar, 2019).
The LWC can be further used to compute the column-integrated liquid water path (LWP,
g/m$^2$) as a function of cloud layer thickness (Δh, meters) (Dong & Mace, 2002; Oh et al., 2018;
Mioche et al., 2017), as follows:

$$LWP = \sum_{\{H_{base}\}}^{\{H_{top}\}} LWC(h).\Delta h \qquad (3)$$

A total of 62 in-situ aircraft profiles (ascending and descending) were constructed from 10135
CDP+2D-S DSD spectra at 1 Hz. From these, in-situ Z, LWC, and LWP were derived and
presented in Fig. 1. The profiles were selected to represent uniform single layer low-level
stratocumulus MBL clouds.

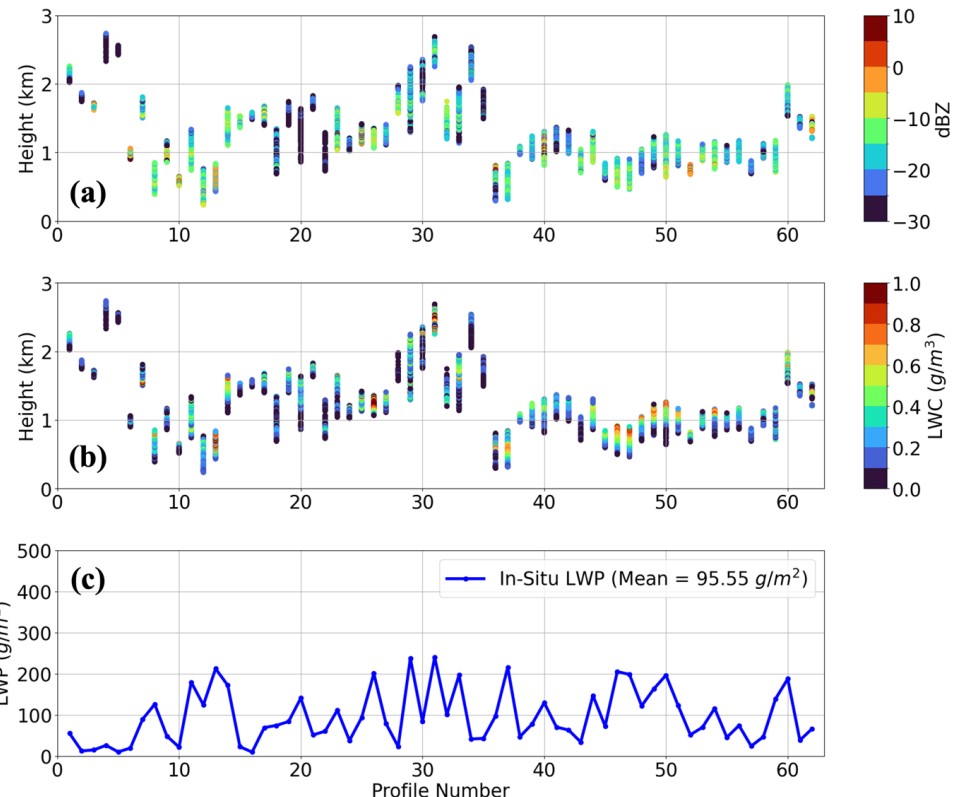

**Figure 1. A total of 62 in-cloud profiles are composed from 10135 (1 Hz) SOCRATES**
**CDP+2D-S DSD spectra to derive in-situ a) reflectivity profiles (Z was converted to dBZ)**
**b) LWC (g/m³) and c) LWP (g/m²) as per equations 1, 2 and 3.**

The in-situ measured LWC and Z can be used to derive an exponential relationship of
form $LWC = aZ^b$, where a, and b are intercept and slope parameters depending on cloud type
(Oh et al., 2018). The in-situ Z and LWC measurements were constrained to only 5[th] to 95[th]
percentile of the dataset to minimize the influence of extreme outliers. Furthermore, a kernel
density estimation (KDE) was used to estimate relative point density in the Z-LWC (log) space.
Due to the noisy nature of the dataset where larger particle diameters return extremely high Z
values ($\sim D^6$) and relatively lower LWC values ($\sim D^3$), a log-log linear regression was performed
only using a subset of the dataset with high sample density to minimize measurement
uncertainties. An exponential relationship: $LWC = 0.70Z^{0.29}$ is hence derived as shown in Fig.
2.




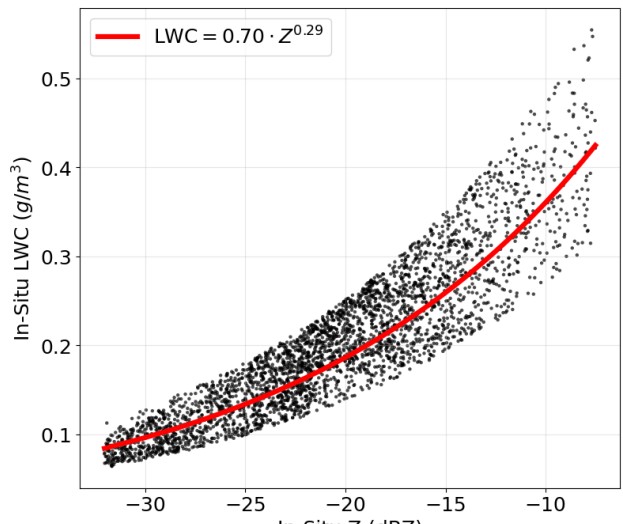

**FIGURE 2. Exponential relationship between Z-LWC derived from the 10135 in-situ**
**DSD measurements for the 62 in-cloud profiles from SOCRATES.**

The predicted LWCs from the LWC-Z relationship are evaluated by against the in-situ
measured LWCs for the 62 profiles of droplet spectra, with means of 0.21 g/m³ for predicted
LWCs and 0.27 g/m³ for in-situ measurements. The root mean square error (RMSE) is
approximately 0.03 g/m³. Furthermore, the predicted LWC profiles were used to estimate
predicted LWPs using equation 3 which and against the 62 in-situ LWPs. The means of
predicted and in-situ LWPs are 93.73 g/m² and 95.55 g/m², with an RMSE around 12 g/m².
The derived LWC-Z relationship, developed following existing studies like Oh (2018)
and Vivekanandan (2020), is specifically tuned to low-level stratocumulus clouds sampled
during the SOCRATES campaign but shall ideally be applicable to similar cloud cases for a
range of W-band reflectivity profiles between -30 to 10 dBZ. Variations in the DSD contribute
to uncertainties in cloud microphysical properties, which could impact the calculated
reflectivity (Z) (Vivekanandan et al., 2020). Additionally, the presence of larger drizzle and
precipitation particles is a major source of uncertainty in LWC-Z power-law relationships, thus
it should be used with caution.
**3 Classifying low-level clouds over SO**
**3.1 Identifying cloud boundaries using HCR measurements**

Existing methods for estimating cloud-base ($H_{base}$) and cloud-top ($H_{top}$) heights, rely on
the thresholds of lidar returned power, depolarization, or backscatter (e.g., Intrieri et al., 2002;
Kang et al., 2021, 2024), as well as in-situ vertically resolved cloud LWP, LWC, and cloud-
droplet number concentration ($N_C$). Kang (2024) presented a method using HSRL backscatter
coefficient ($\beta$), where cloud base is defined as the first lidar range gate where $\beta > 10^{-4}$ m⁻¹sr⁻¹.
This approach is effective when the aircraft is flying below the cloud base with radar-lidar
pointed zenith. Similarly, HSRL signals can accurately detect cloud top when the aircraft is
above the cloud with sensors pointing nadir. However, lidar alone cannot simultaneously
estimate both cloud base and top heights, as HCR does. Therefore, only HCR measurements



were used, which can determine cloud boundaries regardless of the aircraft's altitude or the pointing direction of the remote sensors.

In this study, $H_{base}$ was estimated as the lowest height (from the sea surface) where the HCR Spectrum Width (WID) gradient has the lowest value (or highest negative gradient). The lowest WID gradient indicates the change from a precipitation layer to the cloud layer where the gradient of spectrum width decreases sharply. Higher spectrum width around the cloud base indicates a greater turbulence and wider range of particle velocities observed which correlate to potentially stronger turbulence, and likely drizzle or precipitation. The uncertainties arising from the low signal-to-noise ratios of radar signals near the ocean surface are effectively mitigated because this method only considers the cloud base after eliminating one or two lowest near-surface radar range-gates. $H_{top}$ was estimated as the highest altitude where prominent HCR signals (dBZ > -50) were observed, following the method of Kang (2024). Cloud thickness ($\Delta H$) was then calculated as the difference between cloud-top and cloud-base heights, $\Delta H = H_{top} - H_{base}$. Isolated cloud transects with shallow vertical heights and small horizontal extents were excluded from this study.

After estimating $H_{base}$ and $H_{top}$, cloud types were categorized following the classification method of Xi et al. (2010). A single-layered cloud is defined as having no additional cloud layers above or below during the observed period. Based on this classification, the most common cloud type is found to be single-layered low-level clouds (LOW, $H_{top} \leq 3$ km), followed by middle above low clouds (MOL, $H_{base} < 3$ km, $H_{top} \leq 6$ km) and single-layered middle-level clouds (MID, $H_{base} > 3$ km, $H_{top} \leq 6$ km). Single or double-layered clouds with $H_{base}$ and/or $H_{top} > 6$ km were rarely observed and were excluded to optimize statistical consistency and minimize errors. Only single-layered LOW clouds are considered in this study, which was the predominant cloud type during SOCRATES (~85% of total samples). Furthermore, for cases where cloud vertical columns were intersected by aircraft ascents or descents, HCR cannot reliably observe cloud top or base due to its fixed nadir or zenith-pointing configuration. As a result, only fully sampled cloud profiles—where both cloud base and top are simultaneously observed by the radar—were considered for boundary and phase estimation. Two selected cases of estimated $H_{top}$ and $H_{base}$ for LOW clouds is illustrated in Fig. 3.

## 3.2 Statistical results from prominent cloud types

The derived exponential relationship $LWC = 0.70Z^{0.29}$ was applied to the HCR-observed reflectivity measurements to retrieve a 2D time-height LWC profile and column integrated LWP. Figure 3 illustrates the estimated cloud boundaries, LWC profile and LWP for two selected cases. Figure 3a and 3e clearly shows the derived cloud boundaries and HCR reflectivity profiles. Thicker cloud layers exhibit larger radar reflectivities, which lead to higher estimated cloud LWCs and LWPs as shown in Figs. 3c, 3g, and Figs. 3d, 3h. respectively. LWC profiles basically follow adiabatic increasing from cloud base to upper levels, then decreasing near cloud top due to cloud-top entrainment. As mentioned in Section 2, the derived LWC and LWP values have differences of ~ 22.2% and 1.9%, respectively, compared to aircraft in-situ measurements.



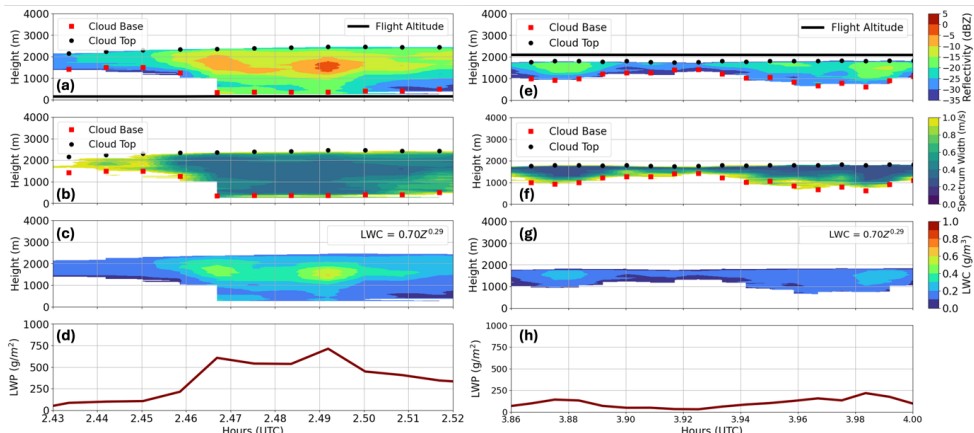

**Figure 3. (a), (b), (c), and (d) are the profiles of HCR reflectivity (dBZ), spectrum width (m/s), LWC (g/m³) and LWP (g/m²), respectively for a selected case from RF1, while (e), (f), (g), (h) are similar profiles but from a selected case from RF7. The identified cloud top (black circles) and cloud base (red squares) heights have been illustrated along the HCR profiles along with the flight altitude (black line). The cases are selected such that the left panels represent the cloud profiles when the aircraft was flying above the cloud top (nadir-pointing radar), while the right panel shows a zenith-pointing radar cross-section with the aircraft flying below the cloud base. The time axis is in decimal hours (UTC).**

LOW clouds were the most frequently observed cloud type—accounting for ~85% of occurrences across the 15 research flights during SOCRATES. In contrast, MID and MOL clouds were less common and together accounted for less than 15%, reflecting the SOCRATES campaign's sampling focus. Some rare occurrences of other high-altitude clouds were also identified but were excluded to maintain statistical consistency. These flights primarily sampled the cold sector of cyclones, occasionally crossing frontal systems associated with strong westerly winds over the Southern Ocean. The combination of large-scale weather patterns and a cool ocean surface led to persistent cloud cover, predominantly low- and mid-level stratus and stratocumulus clouds (McFarquhar et al., 2021; D'Alessandro et al., 2021).

The highest calculated LWP is associated with MOL clouds (~258 g/m²), followed by LOW (~133.2 g/m²) and MID (~48.3 g/m²) clouds, as presented in Table 1. The probability distribution histograms of LWP for the classified LOW, MID, and MOL clouds indicate significantly higher LWP values, with a notable frequency of LOW clouds exhibiting LWPs > 200 g/m². Another distinct peak corresponds to ice clouds with LWPs < 20 g/m². Despite this, overlaps are observed among the probability density functions (PDFs) of LWPs for the different cloud types.

The cloud LWP values during the Measurement of Aerosols, Radiation, and Clouds (MARCUS) ship-based field campaign (DeMott et al., 2018; Mace et al., 2021; McFarquhar et al., 2019, 2021; Marcovecchio et al., 2023; Xi et al., 2022) are retrieved using a physical-iterative algorithm applied to ship-based measurements of microwave radiometer (MWR) brightness temperatures at 23.8 and 31.4 GHz. These retrievals have associated uncertainties ranging from 15 to 30 g/m² (Caddedu et al., 2013), which align with the uncertainty estimates (~20 g/m²) of LWP retrievals obtained via statistical methods (Liljegren et al., 2001). Consequently, a threshold of 20 g/m² is employed in this study to distinguish between liquid clouds (LWP ≥ 20 g/m²) and ice clouds (LWP < 20 g/m²).





**Table 1.** Mean, standard deviation, minimum, and maximum values for estimated cloud base
and top heights, along with calculated LWP for each single-layered cloud type at a 30-second
temporal average.

|  | LOW | MID | MOL |
|---|---|---|---|
| $H_{base} \pm$ **SD**<br>**Min, Max (km)** | 0.89 ± 0.61<br>0.13, 2.95 | 3.97 ± 0.72<br>3.01, 5.7 | 1.79 ± 0.86<br>0.11, 2.95 |
| $H_{top}$ **(mean)** ± **SD**<br>**Min, Max (km)** | 1.70 ± 0.54<br>0.30, 2.99 | 4.44 ± 0.80<br>3.03, 5.91 | 4.10 ± 0.95<br>3.01, 5.91 |
| **LWP (mean)** ± **SD (g/m$^2$)** | 133.17 ± 59 | 48.31 ± 29.4 | 257.95 ± 109 |


## 3.3   Evaluation and comparison of estimated cloud boundaries and LWPs
To evaluate the SOCRATES HCR-estimated boundaries, the $H_{base}$ and $H_{top}$ values were
compared against those estimated by the Micropulse Lidar (MPL), Ceilometer, and 95 GHz
W-band ARM Cloud Radar (WACR) measurements collected during the MARCUS campaign.
These comparisons, conducted for low-level clouds within a spatiotemporally collocated
region over the SO (Section S2, Fig. S1), revealed consistent patterns, as illustrated by the
probability distribution function (PDF) histograms (Figs. 4a and 4b). The mean $H_{base}$ (and $H_{top}$)
height derived from SOCRATES HCR-profiles is ~0.89 km (~1.7 km) with average standard
deviation around 0.55 km. While corresponding estimates from the MARCUS MPL/ceilometer
are around ~0.93 km (~1.41 km) with standard deviation of around 0.58 km. The mean
differences of $H_{base}$ and $H_{top}$ are ~0.04 km and ~0.29 km, respectively. Furthermore, HCR-
derived $H_{base}$ values were compared with those derived from GV-HSRL lidar observations for
periods where the aircraft flew below the cloud base with a zenith-pointing radar-lidar view
direction. The HSRL-detected $H_{base}$ was identified as the first range gate where the backscatter
coefficient (β) exceeded $10^{-4}$ m$^{-1}$sr$^{-1}$ (Kang et al., 2024).  In these cases, which comprised
~20% of the total samples, both HCR and HSRL simultaneously detected cloud bases. A
statistical comparison revealed a strong correlation of 0.78 (p << 0.001) and RMSE of 0.29 km
(Fig. 4c). Notably, HSRL-retrieved $H_{base}$ heights are consistently higher than those derived
from HCR, by approximately 300-400 m.
Figure 4d presents PDFs comparing SOCRATES-derived LWP with MWR-retrieved
LWP from the MARCUS campaign for LOW clouds, showing similar trends between the two
datasets. The mean LWP derived from SOCRATES low clouds is 133.2 g/m$^2$, while MARCUS
MWR yields a mean of 123.96 g/m$^2$. Overall, the statistical comparisons of $H_{base}$, $H_{top}$, and
LWP for LOW clouds derived from the SOCRATES aircraft-based remote sensors
demonstrate decent agreement with corresponding estimates from ship-based measurements
during the MARCUS campaign, within the measurement uncertainties. The differences in
observed measurements arise from the distinct (spatiotemporal) sampling strategies of the two
campaigns which makes it difficult to obtain precisely overlapping cloud profile observations.





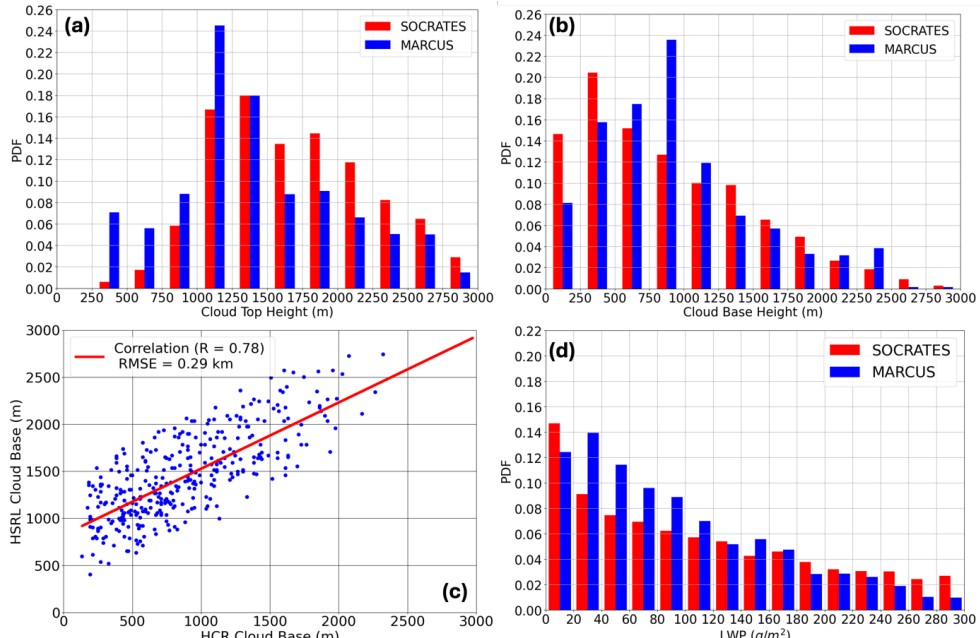

**FIGURE 4. Comparison of the estimated cloud boundaries and LWP: Probability Density Function (PDF) histograms for (a) cloud base heights, and (b) cloud top heights derived from SOCRATES HCR- and MARCUS MPL/Ceilometer/WACR- measurements. (c) Cloud-base heights derived from SOCRATES HCR and HSRL measurements when the aircraft flew below the cloud base for all 15 research flights. (d) LWP comparison between SOCRATES and MARCUS MWR retrievals.4 Low-level cloud phase classification method and discussion**

As discussed previously, LOW clouds are the dominant cloud type (~85%) observed during SOCRATES. This section outlines the method for determining LOW cloud phase.

### 4 Low-level cloud phase estimation

As discussed previously, LOW clouds are the dominant cloud type (~85%) observed during SOCRATES. This section outlines the method for determining LOW cloud phase. To ensure reliable phase classification, we exclude segments prone to high uncertainty, such as noisy pixels or very thin layers from steep sawtooth crossings and include only complete cloud profiles (as mentioned in Section 3.1).

### 4.1 Determination of cloud phase

Figure 5 presents the flowchart for determining cloud phase for the classified LOW clouds ($H_{top}$ < 3 km) after applying the in-cloud condition constraint (LWC > 0.01 g/m³). The phase partitioning method is used as a set of combined filters, classifying cloud phase as a 2D profile of liquid, mixed, and ice phases. Additional cloud phase type classes, such as drizzle, rain, and snow are also classified. Phase classification is carried in a stepwise manner. The cloud phase classification in this study—based on profiles of air temperature (T), LWP, HCR reflectivity (dBZ), Doppler velocity ($V_d$), and Doppler spectrum width (WID), as described in Fig. 5—is



performed in conjunction with the bivariate histograms presented in Fig. 6 for classifying
liquid, mixed, ice, drizzle, rain and snow phase types. Because of overlapping constraints
across multiple datasets, preserving the order of classification is important.

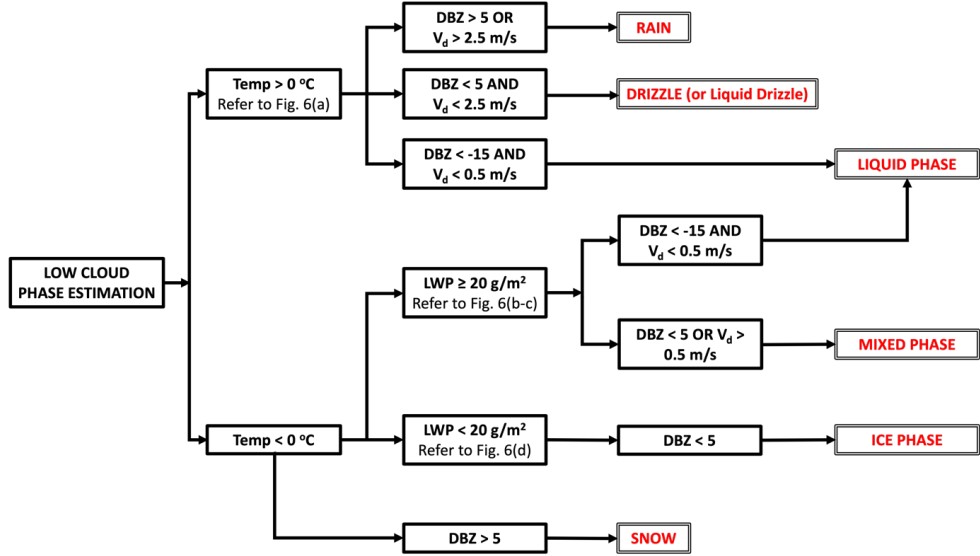


**Figure 5. Flow chart depicting the phase classification of single-layered LOW clouds
during SOCRATES. Temperature is provided from ERA5 reanalysis air temperature
product.**

Regions of strong precipitation are identified based on cloud profiles exhibiting strong
HCR reflectivity (dBZ > 5), classified as rain in warm conditions (temperature, T > 0 ℃) or
snow in cold conditions (T < 0 ℃). Additionally, warm cloud regions (T > 0 ℃) with extremely
strong downdrafts (doppler velocity, $V_d$ > 2.5 m/s) are also categorized as rain. Clouds with T
> 0°C, dBZ < -15, and weak updraft ($V_d$ < 0.5 m/s) are classified as liquid while drizzle
typically has higher reflectivity (> -15 dBZ, Wu et al., 2020a) and $V_d$ < 2.5 m/s (moderately
strong downdraft/turbulence). Figure 6a presents the classification of hydrometeor types within
warm cloud regions, distinguishing between liquid-phase clouds (dBZ ≤ 15, $V_d$ < 0.5 m/s),
drizzle (-15 < dBZ < 5 or $V_d$ > 0.5 m/s), and rain (dBZ > 5). A clear linear relationship between
$V_d$ and dBZ is observed, with drizzle dominating the distribution, even in cases where dBZ <
-15 and $V_d$ > 0.5 m/s. These findings are consistent with Marcovecchio (2023), who reported
a high drizzle frequency (71.8%) in MARCUS field campaign data.
As discussed in Section 3.2, a threshold of LWP = 20 g/m² is used to classify cloud
phase: ice clouds (LWP < 20 g/m²) and mixed-phase or liquid clouds (LWP ≥ 20 g/m²), as
illustrated in Fig. 5. The width of the Doppler spectrum (WID) serves as an indicator of cloud
microphysical variability, with lower WID values suggesting homogeneous, single-phase
clouds, and higher WID values indicating increased turbulence, wind shear, or mixed-phase
conditions (Shupe, 2007). In subfreezing regions (T < 0°C), clouds characterized by low WID,
and weak updrafts ($V_d$ < 0.5 m/s) are classified as liquid, typically composed of small droplets
and supercooled liquid water (SLW). Mixed-phase clouds are identified when both WID and
$V_d$ exceed 0.5 m/s, indicating greater turbulence and broader hydrometeor size distributions.
Clouds with WID > 0.5 m/s and $V_d$ < 0.5 m/s (or vice versa) are reclassified based on
reflectivity: as mixed-phase when dBZ > –15, and as liquid when dBZ < –15. Since radar





reflectivity is proportional to the sixth power of particle diameter (Wang et al., 2009), clouds composed of small, uniform liquid droplets exhibit lower dBZ values, while mixed-phase clouds produce higher reflectivities due to the presence of larger particles. The classification of liquid and mixed-phase clouds under varying turbulence conditions is shown in Fig. 6b (WID > 0.5 m/s) and Fig. 6c (WID < 0.5 m/s). Most reflectivities fall below −15 dBZ, and a $V_d$–dBZ linear relationship, similar to that in Fig. 6a but with different slopes, is observed. The 2D distribution pattern in Fig. 6c resembles that of Fig. 6a, further supporting the dominance of drizzle in LOW clouds over the SO. Ice-phase regions (T < 0°C, LWP < 20 g/m²) are depicted in Fig. 6d, where ice is distinguished from snow in low-turbulence environments (WID < 0.5 m/s) by low reflectivity (dBZ < 5).

In regions where LWP ≥ 20 g/m², pixels with elevated LWCs (> 0.2 g/m³) are reclassified as pure liquid-phase clouds. Snow classification, presented in Figs. 6b–d, applies to all regions with T < 0°C. Where both LWP > 20 g/m² and subfreezing temperatures occur, precipitation may include supercooled (or freezing) rain in turbulent environments (WID > 0.5 m/s, Fig. 6b), or rimed snow and graupel in less turbulent settings (WID < 0.5 m/s, Fig. 6c). These regions are uniformly identified as snow, based on the assumption that supercooled precipitation freezes upon descent due to contact with airborne particles in downdrafts (Brownscombe & Hallett, 1967).

Although the phase-classification thresholds for WID, $V_d$, and dBZ were specifically tuned for clouds sampled during SOCRATES, we expect them to be broadly applicable to MBL clouds over the SO. To ensure consistency, the phase-diagnostic thresholds adopted in this study were compared with values reported in previous studies (e.g., Xi et al., 2022; Romatschke & Vivekanandan, 2022; Desai et al., 2023; Shupe, 2007). As Shupe (2007) noted, there are occasional cases where the applied $V_d$ and WID thresholds may suggest a mixed or liquid phase even though ice is the dominant hydrometeor type at that altitude.

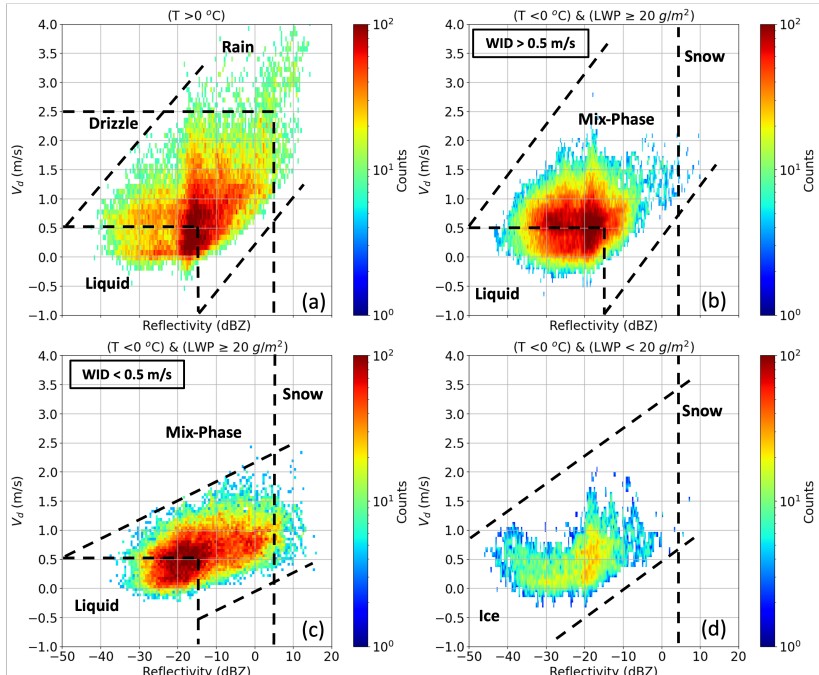

**Figure 6. (a-d) Bivariate histograms of HCR reflectivity (dBZ) and Doppler velocity ($V_d$) for different spectrum widths (WID), LWP and temperatures (T) to demonstrate the classified liquid, ice and mixed phase cloud samples. The colorbar shows the sample count in each bin in a log-scale and the dashed lines represent the threshold values for the phase classification. The phase class types of rain, drizzle, and snow are also classified.**

## 4.2 Results for LOW cloud phase classification

An LWP greater than the retrieval uncertainty ($\geq 20$ g/m$^2$) indicates the presence of liquid water (at T> 0 ºC) or supercooled liquid water (SLW, at T < 0 ºC). Larger ice particles, being denser than liquid droplets, typically exhibiting broader Doppler spectrum widths and higher fall speeds (Xi et al., 2022). A significant number of drizzle (at T > 0 ºC) and mixed-phase (T < 0 ºC) samples were observed near cloud base, likely driven by elevated WID and $V_d$ values. Mixed-phase clouds represent a complex three-phased colloidal system in which SLW droplets coexist with ice crystals, influencing both the nature of mixed-phase layers (genuinely and/or conditionally mixed) and underlying microphysical processes (Korelov & Milbrandt, 2022; Maciel et al., 2024). D'Alessandro (2021) found that SO clouds sampled during SOCRATES exhibited significant spatial heterogeneity. Low-level clouds generally exhibit higher temperatures than recorded aircraft temperatures due to altitude differences between flight paths and cloud boundaries. Analyzing the ERA5 air temperature indicates that mode temperatures ranged between -5°C and 0°C for all three phases (liquid, mixed and ice). Notably, mixed-phase clouds show the highest occurrence between -15°C to -2.5°C, underscoring the spatial heterogeneity of low-level stratocumulus clouds, consistent with the findings from D'Alessandro (2021) and Maciel (2024).

Based on a 30-second temporal averaging interval, the classification method developed in this study identified liquid-phase clouds, including SLW, as the most frequent category, accounting for 48.79% of all samples. Mixed-phase and ice-phase clouds comprised 23.26% and 6.88%, respectively. Increasing the temporal averaging interval results in a higher



proportion of mixed-phase clouds and a corresponding decrease in the occurrence of single-
phase clouds. Drizzle was identified in 16.22% of the cases, while rain (3.35%) and snow
(1.51%) were relatively rare. Most of the rain and snow detections correspond to in-cloud
precipitation, with some falling hydrometeors observed near cloud base.
Figure. 7a–e illustrates the profiles for HCR-dBZ, WID, $V_d$, LWC and LWP for a flight
case (RF13), with Fig. 7f showing the resulting classified phases. Notably, regions where
mixed-phase layers overlay liquid-only columns suggest cloud-top entrainment. In these cases,
mixing of dry air into the cloud top enhances evaporation of liquid droplets, promoting the
formation of mixed-phase conditions. Analysis of 2D-S particle probe imagery (Wu and
McFarquhar, 2019) reveals that liquid droplets are predominantly spherical, whereas large ice
particles display irregular morphologies. While large ice particles ($D_p > 200$ μm) can be
visually identified by shape, small ice crystals are not well-resolved by the 2D-S probe, making
it difficult to distinguish them from similarly sized cloud droplets.

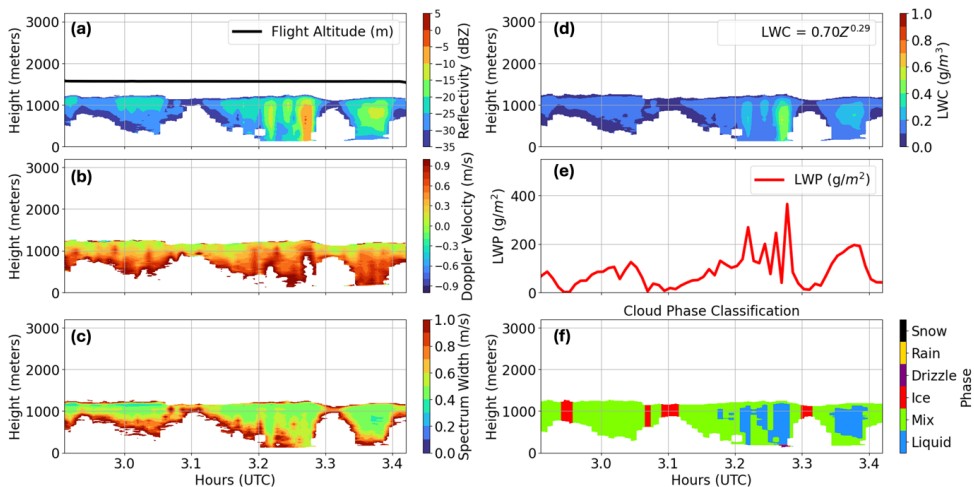


**Figure 7. A case study from SOCRATES flight RF13 illustrating the phase-classification
algorithm. Shown are: (a) HCR reflectivity (dBZ) with the flight altitude (black line), (b)
Doppler velocity ($V_d$), and (c) Doppler spectrum width (WID) profiles. (d) LWC profile
(retrieved from HCR reflectivity), (e) column-integrated LWP, and (f) classified cloud
phase categories. The time axis is in decimal hours (UTC).**

**4.3 Evaluation of phase-classification with existing methods**
Cloud phase classification is highly sensitive to observational scales, sampling strategy, and
instrumentation type. This study focuses exclusively on low-level SO clouds observed during
SOCRATES. The phase-classification method developed here (hereinafter referred to as HCR-
phase) is evaluated through comparisons with five existing methodologies, grouped into three
categories: 1) machine learning methods based on in-situ probes — this includes multinomial
logistic regression (MLR) (D'Alessandro et al., 2021; Schima et al. 2022) and the University
of Washington Ice–Liquid Discriminator (UWILD) (Atlas et al., 2021); 2) ship-based radar-
lidar-MWR measurements from MARCUS (Xi et al., 2022; Zhang and Levin, 2017) ; 3) fuzzy
logic particle identification (PID) — developed from airborne HCR and HRSL measurements
during SOCRATES (Romatschke and Vivekanandan, 2022). Only low-level cloud samples
below 3km were used for cross-method comparisons. The comparative analysis aims to assess





the strengths and limitations of the HCR-phase classification in the context of these existing
methodologies.

### 4.3.1 Comparison with in-situ phase classification (machine learning methods: MLR and UWILD)

D'Alessandro (2021) developed a cloud phase classification method using an MLR
model trained on in-situ measurements from the CDP, 2D-S, and Rosemount Icing Detector
(RICE) collected during the SOCRATES campaign. This model classified cloud phase—
liquid, mixed, or ice—for samples at air temperatures below 0°C. Schima (2022) refined the
MLR product by manually evaluating imagery from the 2D-S, 2D-C, and PHIPS (Particle
Habit Imaging and Polar Scattering) probes. Among 1600 in-situ samples collected below 3
km altitude, the MLR approach classified 52.25% as liquid, 9.5% as mixed-phase, and 38.2%
as ice clouds. Approximately, 39% of samples at temperatures above freezing (T > 0°C) remain
unclassified; including these would likely raise the overall fractions of liquid and mixed-phase
clouds in the MLR dataset. Compared to the HCR-phase method developed in this study, the
MLR model detects a larger fraction of liquid clouds but underrepresents mixed-phase and ice
clouds.
HCR-phase classification was also compared with the UWILD product, which uses a
random forest algorithm trained on 2D-S probe images and particle inter-arrival times (Atlas
et al., 2021; Mohrmann et al., 2021), and validated with PHIPS imagery. For consistency,
UWILD timestamps with nearly equal liquid and ice samples were reclassified as mixed phase.
Among time periods overlapping with HCR-phase (matched to the nearest second), UWILD
classified 58.8% of samples as liquid, 38.9% as mixed-phase clouds, and only 2.3% as ice-
phase clouds. Increasing the temporal averaging interval led to a higher frequency of mixed-
phase classification (77%), with reduced occurrences of liquid (19%) and ice (4%). A key
source of uncertainty arises from the spatial heterogeneity of clouds and the UWILD
algorithm's lack of an explicit mixed-phase category. Additionally, UWILD defines all small
particles (equivalent diameter $D_{eq} < 0.17$ mm) as liquid, which likely underrepresents ice-phase
clouds. This limitation is significant, as the SOCRATES campaign sampled very few particles
larger than 0.17 mm, even within clouds confirmed to be in the ice phase (Zheng et al., 2024;
Wu and McFarquhar, 2019).
Within the same samples, the HCR-phase method classified 61.81% of samples as
liquid, 29.47% as mixed-phase clouds, and 8.71% as ice-phase clouds. A key challenge in
comparing in-situ ML-based phase classifications with the HCR-phase method lies in the
differing observational frameworks. Both the MLR and UWILD classifiers are trained on
microphysical probe data (e.g., CDP and 2D-S) collected at the GV aircraft's altitude, and
therefore cannot provide phase information across the full vertical extent of HCR-observed
cloud profiles. This limitation is especially pertinent when the aircraft is flying above or below
cloud layers. Furthermore, the MLR algorithm is specifically designed for classifying cloud
phases under subfreezing conditions, making it most applicable to cold cloud environments. A
substantial number of unclassified points—particularly those at temperatures above 0°C—are
excluded from final phase statistics, further complicating direct, one-to-one comparisons.
Nevertheless, this comparison does provide a valuable approximation of how in-situ measured
cloud phases compares to HCR-derived cloud phases.

### 4.3.2 Comparison with ship-based phase classification during MARCUS

The HCR-phase classification was further evaluated through comparison with single-
layer, low-cloud (<3 km) phase classifications from the ship-based measurements during the



MARCUS campaign, focusing on a spatiotemporally collocated region over the SO using a 5-
minute averaging interval (see Section S2). Zhang and Levin (2017) developed the
thermodynamic cloud phase product (THERMOCLDPHASE; ARM, 2017) for MARCUS
using data from the U.S. DOE's ARM Mobile Facility deployed aboard the *Aurora Australis*.
This product integrates active remote sensing instruments—including the Micropulse Lidar
(MPL) and W-band ARM Cloud Radar (WACR)—with passive sensors such as microwave
radiometers (MWR) and radiosondes. Cloud phase classification is based on the Shupe (2005,
2007) methodology and includes seven thermodynamic hydrometeor types. More recently, Xi
(2022) introduced an improved classification approach tailored to single-layer low-level clouds
over the SO, leveraging WACR Doppler spectra, MWR-derived LWP, and radiosonde-based
temperature profiles.
For a total of 1410 5-min samples, Xi (2022) classified 58.6% as liquid-phase, 30.7%
as mixed-phase, and 10.6% as ice-phase clouds. In contrast, the thermodynamic cloud phase
product (ARM, 2020) identified 52.31% liquid, 25.15% mixed, and 22.53% ice-phase clouds
across 3435 samples. The statistical results from both these methods remain broadly consistent
within the same phase categories with those obtained using the HCR-phase classification
(61.81% liquid, 29.47% mixed-phase, and 8.71% ice; excluding drizzle, rain, and snow). The
differences in the phase partitioning are expected, given the inherent limitations in achieving
perfect spatiotemporal alignment between the SOCRATES and MARCUS campaigns.
**4.3.3 Comparisons with fuzzy logic particle identification (PID)**

Romatschke and Vivekanandan (2022) developed a technique for identifying
hydrometeor particle types using HCR and HSRL observations during the SOCRATES
campaign. A fuzzy logic particle identification (PID) algorithm was developed using HCR,
HSRL, and temperature parameters (NCAR/EOL HCR Team, 2023) to classify 11 distinct
hydrometeor types. This PID dataset served as a valuable reference for validating the HCR-
phase classification through a pixel-by-pixel comparison for low-level clouds, using a 10-
second temporal average. To ensure consistent comparisons, the PID-classified 'cloud liquid'
and 'supercooled cloud liquid' categories were merged into a single 'cloud liquid' category,
while categories such as 'supercooled drizzle' and 'supercooled rain' were excluded due to
their minimal sample counts. Table 2 summarizes the raw sample (pixel) counts for each phase
category, comparing results between the HCR-phase classification and the PID-derived
hydrometeor types.
**Table 2. Comparison of HCR-phase classification and PID at a 30-second temporal**
**resolution.**

| Numbers represent pixel counts | | PID Scheme | | | | | | | |
|---|---|---|---|---|---|---|---|---|---|
| | | Cloud | Precipitation | Small Frozen | Large Frozen | Melting | Cloud Liquid | Drizzle | Rain |
| HCR-Phase | Liquid | 54114 | 563 | 4135 | 760 | 178 | 20084 | 6 | 0 |
| | Mix | 19448 | 182 | 754 | 18 | 2 | 13784 | 0 | 0 |
| | Ice | 8039 | 59 | 270 | 33 | 2 | 2902 | 0 | 0 |
| | Drizzle | 2115 | 25 | 149 | 40 | 590 | 3614 | 17814 | 0 |
| | Rain | 245 | 3 | 0 | 88 | 614 | 589 | 2612 | 1326 |
| | Snow | 22 | 30 | 68 | 2115 | 31 | 6 | 5 | 0 |


Figure 8 illustrates the distribution of PID-scheme phase categories corresponding to
each HCR-phase, based on overlapping pixels where both methods provide valid results. For





these matched samples, the HCR-phase classification identifies cloud and in-cloud
precipitation hydrometeor types as approximately 64.4% (65.3%) liquid, 23.2% (21.1%)
mixed-phase, and 9.6% (6.8%) ice-phase. Notably, 69.3% of the large frozen hydrometeor
types from the PID are classified as snow in the HCR-phase, reflecting the threshold-based
classification scheme used in this study, which tends to identify large ice particles with high
reflectivity (dBZ) and low LWP as snow. Similarly, PID-classified drizzle is mapped to 87.2%
drizzle and 12.8% rain in HCR-phase, while all PID-classified rain is consistently identified as
rain by HCR-phase. The melting layer observed in MBL clouds, characterized by enhanced
dBZ values similar to drizzle, contains both liquid and ice-phase particles as hydrometeors melt
while falling through the 0°C isotherm (Song et al., 2021). Accordingly, HCR-phase classifies
PID melting layer samples as 41.6% drizzle, 43.3% in-cloud rain, and 12.6% liquid clouds.
The greatest mismatch occurs for PID-classified small frozen hydrometeors, which are
identified as 76.9% liquid, 14% mixed-phase, and only 5% ice by the HCR-phase method. This
discrepancy likely reflects the presence of supercooled liquid droplets coexisting with ice
particles, which HCR-phase identifies using its criteria of low dBZ, $V_d$, and high LWP; while
the PID algorithm do not use any LWP constraints. Among the matched samples, PID-
classified 'cloud liquid' (merged with 'supercooled cloud liquid') types account for 49% of
HCR-phase liquid and 33.6% of mixed-phase classifications. These percentages are derived
from the subset of low-level clouds with collocated pixels and do not represent the full PID
dataset.

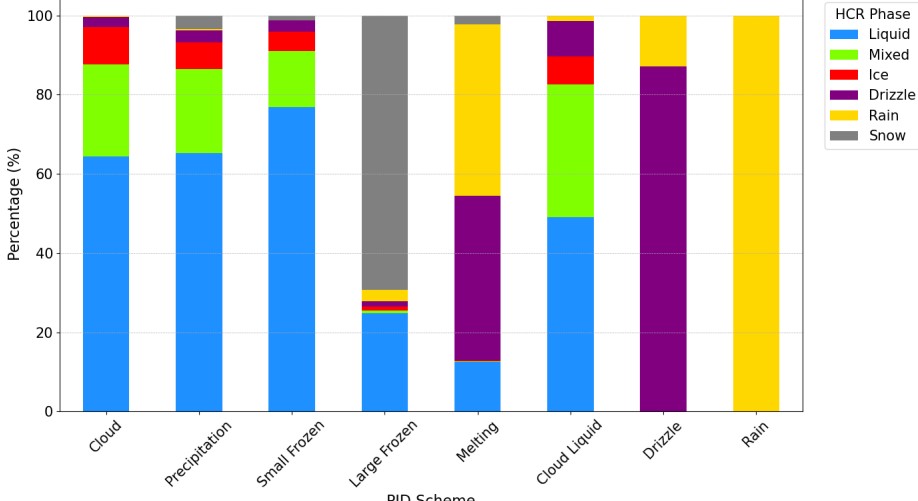

**Figure 8. The distribution of phase partitioning of hydrometeor types identified by the**
**PID algorithm, showing the percentages of each phase-category samples as per the**
**overlapping pixels with HCR-phase.**
**4.4 Summarizing the comparisons between HCR-phase and other methods**
Table 3(a,b) summarizes the low-level cloud phase classifications from the HCR-phase
algorithm alongside results from five other methods. For the SOCRATES campaign,
comparisons include the MLR method (Schima, 2022; D'Alessandro, 2021, 2022), and the
fuzzy logic PID algorithm (Romatschke and Vivekanandan, 2022). For the MARCUS
campaign, comparisons are made with WACR-MWR retrievals (Xi et al., 2022) and the
Thermodynamic Cloud Phase product (ARM, 2017).





The MLR method predicts 9.6% fewer liquid-phase clouds, 20% fewer mixed-phase, and 29.5% fewer ice-phase clouds compared to the HCR-phase (Table 3a, column 2). Additionally, ~39% of the MLR samples are unclassified in regions exceeding 0°C; which were excluded from the comparison. In contrast, UWILD results show closer agreement with HCR-phase, with only a 3% difference in liquid-phase occurrence. However, UWILD reports 9.4% more mixed-phase and 6.4% fewer ice-phase clouds (Table 3a, column 3). For the MARCUS campaign, the Xi (2022) phase classification method aligns closely with HCR-phase, underestimating liquid-phase clouds by just 3.2% and differing by only 1.2% and 1.9% in mixed and ice phases, respectively (Table 3a, column 5). Similarly, the Thermodynamic Cloud Phase product identifies 7.3% more ice-phase clouds, but 2.3% fewer liquid and 5% fewer mixed-phase clouds relative to HCR-phase. Table 3b compares the HCR-phase results with the fuzzy logic PID scheme. The PID method classifies 56.5% of matched low-level samples as liquid (including supercooled liquid) and 11.6% as frozen (large and small frozen hydrometeors), which is 7.7% higher in liquid and 3.2% higher in frozen phase (ice and snow samples) occurrence than HCR-phase. Additionally, PID estimates 11.9% more drizzle and 1.52% less rain compared to HCR-phase, highlighting notable differences in hydrometeor subtype identification.

Overall, the phase classification percentages show reasonable agreement across all methods, with the MLR method displaying the largest deviations. Mixed-phase cloud identification remains the most uncertain, particularly in regions with high spatial heterogeneity. The HCR-phase method, which combines radar observations with in-situ measurements, demonstrates strong capability in detecting mixed-phase clouds due to two primary factors: (1) the integrated use of HCR reflectivity (dBZ), spectral width (WID), and Doppler velocity ($V_d$) effectively characterizes particle size distributions, enabling clear differentiation between mixed-phase, drizzle, liquid, and ice clouds; and (2) the use of a 20 g/m² liquid water path (LWP) threshold helps distinguish ice-dominated columns from liquid and mixed-phase cloud regions.

**Table 3. (a) Comparison of HCR-phase classification with in-situ ML phase products (MLR, UWILD) and MARCUS-phase retrievals. For HCR-phase, drizzle, rain, and snow categories are excluded from this comparison. (b) Comparison of HCR-phase with PID scheme. Frozen categories combine ice and snow (for HCR-phase), large frozen and small frozen (for PID).**

| (a) | Results from SOCRATES | | | Results from MARCUS | |
|---|---|---|---|---|---|
| | HCR-Phase | MLR | UWILD | WACR-MWR | Thermo-cloud phase |
| Liquid % | 61.81 | 52.25 | 58.8 | 58.65 | 56.82 |
| Mix % | 29.47 | 9.50 | 38.9 | 30.71 | 27.2 |
| Ice % | 8.71 | 38.25 | 2.3 | 10.64 | 15.98 |

| (b) | HCR-Phase | PID-scheme |
|---|---|---|
| Liquid % | 48.79 | 56.45 |
| Mix % | 23.26 | - |
| Melting % | - | 1.95 |
| Frozen % | 8.39 | 11.61 |
| Drizzle % | 16.22 | 28.15 |
| Rain % | 3.35 | 1.83 |





The percentage agreement for each phase is calculated by comparing the results from
this study with those from the other methods. Agreement is defined as:
**Agreement (%) = 100 - |(Phase$_i$% from compared method) - (Phase$_i$% from HCR-phase)|,**
where i represents the phase categories.
Results show that the MLR method has the lowest agreement with HCR-phase for the
mixed (80%) and ice (70.5%) phase categories, although it performs well for liquid clouds
(90.4%). In contrast, the UWILD, WACR-MWR, and thermodynamic cloud phase methods
exhibit strong agreement with HCR-phase, with >90% consistency across all three phase
categories. The PID scheme also shows high agreement (~92.3%) with HCR-phase for liquid,
frozen, drizzle, and rain categories. As previously discussed, the underlying differences
between these classification methods - in terms of sensor type, sampling strategy, and
algorithm design - contribute to the observed deviations. While direct one-to-one comparisons
are inherently limited by methodological differences, these analyses offer a robust foundation
for evaluating phase classification accuracy and consistency across platforms.
**5 Summary and Conclusions**

This study presents a comprehensive methodology for identifying cloud boundaries and
classifying cloud phases in single-layer low-level marine boundary layer (MBL) clouds. The
approach leverages data from the airborne HCR radar and in-situ cloud and drizzle probes
(CDP and 2D-S) aboard the NSF/NCAR GV aircraft during SOCRATES campaign over the
SO. Cloud base detection is achieved using the Doppler spectrum width gradient, which
reliably identifies cloud boundaries even in the absence of supporting sonde or ceilometer
measurements. The cloud phase classification (referred as HCR-phase), integrates radar
observations with in-situ measurements, enabling accurate phase retrieval in the time-height
dimension. In addition, the study provides some new insights into the macrophysical properties
of different cloud types and phases through statistical analyses. Both the cloud base detection
and phase classification methodologies were rigorously evaluated against existing methods.
The major takeaways from this study are as follows:

1. A method based on HCR-reflectivity and spectrum width gradient-based method was
used to identify cloud boundaries and classify cloud types into LOW, MID, and MOL
categories, based on cloud-top and cloud-base heights. LOW clouds (< 3km) were the
most frequently observed, occurring in 85% of observed cases. HCR- and HSRL-
derived cloud base heights showed strong agreement, with a correlation of 0.78. PDFs
of cloud base and top heights from SOCRATES (HCR) and MARCUS
(MPL/ceilometer/WACR) over a collocated region also exhibited close agreement,
with mean differences of 0.04 km for cloud base and 0.29 km for cloud top.
2. In-situ measured LWC and calculated reflectivity (Z) from CDP and 2D-S probe
measurements were used to derive an empirical exponential relationship: LWC =
$0.70Z^{0.29}$. This relationship was applied to HCR-reflectivity data to retrieve vertical
LWC profiles over time. Using these LWC profiles and cloud thickness, LWP was
estimated for each cloud category, with an associated uncertainty of approximately
±20 g/m². The mean LWP values were 133.2 g/m² for LOW clouds, 258 g/m² for MOL
clouds, and 48.3 g/m² for MID clouds. Additionally, LWP estimates for SOCRATES
low clouds were in close agreement with microwave radiometer (MWR)–retrieved
LWP values from MARCUS, showing a mean difference of just 9.24 g/m².
3. A phase classification method (HCR-phase) was developed to categorize single-layered
low-level (LOW) clouds into liquid, mixed, and ice phases, with respective occurrence
frequencies of 48.8%, 23.3%, and 6.9%. Additional hydrometeor types, including



drizzle (16.2%), rain (3.4%), and snow (1.5%), were also identified. The HCR-phase
classifications were benchmarked against five existing methods: MLR, UWILD, PID,
WACR-MWR, and Thermodynamic Cloud Phase. Overall, HCR-phase showed strong
agreement (>90%) with these methods across the primary phase categories. The largest
deviation was observed with the MLR method, which showed lower agreement (~70%)
for mixed and ice phases, primarily due to methodological differences in classification
criteria.
This study advances our understanding of Southern Ocean clouds by introducing robust
methodologies for identifying cloud boundaries and classifying cloud phases in MBL clouds.
The techniques developed here are broadly applicable to future field campaigns and research
efforts aimed at characterizing MBL cloud properties in other remote maritime regions. A
promising avenue for future investigation lies in the integration of HCR and HSRL
observations which has significant potential in improving cloud phase and microphysical
property retrievals. Further work may focus on refining the cloud boundary and phase
classification algorithms, incorporating additional remote sensing instruments, and assessing
the implications of cloud phase heterogeneity on aerosol–cloud–radiation interactions.
**Data Availability.** All radar-lidar and in-situ data from the NSF SOCRATES campaign used
in this study are freely available via the EOL data archive (https://data.eol.ucar.edu/dataset/)
and the SOCRATES website (https://www.eol.ucar.edu/field_projects/socrates). The 2D-S
dataset (Wu & McFarquhar, 2019) is available at https://doi.org/10.26023/8HMG-WQP3-
XA0X, and CDP data (UCAR/NCAR-EOL, 2022) at https://doi.org/10.5065/D6M32TM9. The
MLR cloud phase dataset (D'Alessandro et al., 2021) can be found at
https://doi.org/10.26023/S6WS-G5QE-H113 (https://data.eol.ucar.edu/dataset/552.142), and
UWILD phase classification (Mohrmann et al., 2021) at https://doi.org/10.26023/PA5W-
4DRX-W50A (https://data.eol.ucar.edu/dataset/552.134). NCAR HCR radar and GV-HSRL
lidar moments data (NCAR/EOL HCR Team, NCAR/EOL GV-HSRL Team, 2023), including
the fuzzy logic PID scheme (Romatschke & Vivekanandan, 2022), are available at
https://doi.org/10.5065/D64J0CZS (https://data.eol.ucar.edu/dataset/552.034). Additional
datasets, including ARM MWRRET1LILJCLOU, ARSCLWACRBND1KOLLIASSHP, and
THERMOCLDPHASE for the MARCUS campaign, can be accessed via the ARM data store
at https://adc.arm.gov/.
**Author contributions**. The idea of this study was discussed by AD, BX, and XD. AD
performed the analyses and wrote the paper. AD, BX, XD, and XZ participated in the scientific
discussions and provided substantial comments and edits on the paper.
**Competing interests**. The contact author has declared that neither they nor their co-authors
have any competing interests.
**Acknowledgments.** The SOCRATES aircraft dataset, campaign details, and related
publications are freely available at https://www.eol.ucar.edu/field_projects/socrates. We thank
John D'Alessandro (University of Washington) for guidance on the MLR phase determination
method, Ulrike Romatschke (NCAR) for explaining radar-lidar fuzzy logic parameters, and
Christopher J. Webster (NCAR) for assistance with XPMS2D software. This manuscript was
proofread using ChatGPT and Grammarly.
**Financial Support.** This work was supported by the University of Arizona's IT4IR TRIF and
Provost Investigation funds. The researchers at the University of Arizona were also supported
by NSF grant AGS-2031750 at the University of Arizona. The work at Argonne National

5000



Laboratory was supported by the U.S. DOE Office of Science under contract DE-AC02-
06CH11357.

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
