# Peer review of "Identifying MBL cloud boundaries and phase over the Southern Ocean using airborne"

_EGUsphere, 2025_

## Author Comment (AC1)

**Response to Reviewers,**

We thank and appreciate the Reviewers and the EGU AMT Editorial team for their valuable scrutiny and feedback on our manuscript (**egusphere-2025-874**) and the opportunity to resubmit a revised version. All the suggested changes and feedback have been duly addressed and implemented to the best possible extent, with our responses for each comment presented as follows. Please note that this manuscript has already been subject to previous rounds of reviewer comments and editorial suggestions, which were already implemented in previous rounds of resubmission.

**Major changes in the revised manuscript are as follows:**

1. Analysis is now restricted to low-cloud cases, with all references to MID and MOL cloud types removed.
2. **Cloud-base identification** – We feel the cloud-base estimation method is an important part of this paper and also for cloud phase section. Therefore, we want to keep this section but the method has been completely revised via using an array of HCR measurements for detecting cloud base in both drizzling and non-drizzling cases, evaluated on a case-by-case basis with HSRL lidar and aircraft in-situ measurements during SOCRATES field campaign
3. The revised cloud-base detection approach has been extensively validated against lidar- and in-situ-derived cloud bases. Table 1 and Figure 3 have been updated to illustrate HCR-derived cloud bases and their comparison with HSRL-based estimates for drizzling and non-drizzling cases.
4. **Phase-classification results** - following the changes in cloud boundary estimation, the final phase results derived from the phase detection method have also changed significantly. Figure 5 represents the new phase-detection algorithm tuned specifically for drizzling and non-drizzling cases, along with tuning for the regions above- (in cloud) and below-cloud base.
5. **Phase classification comparison** – A more targeted pixel-by-pixel comparison between the HCR-phase method and the probe-based MLR-phase product has been added, following Reviewer 2's suggestion. This comparison uses only fully sampled cloud profiles with both cloud top and base observed, yielding 298 nearest-overlapping samples within 100–200 m. These sample limitations contribute to statistical variability as only fully sampled cloud profiles, where both cloud top and base were observed simultaneously, were utilized for phase classification (this excludes all sawtooth transects). Further, comparisons with the UWILD method have been removed, as it provides no substantial new information beyond the MLR product.
6. **Revised tables and figures** – Table 2 now reports only hit rates (agreement percentages) and counts of overlapping samples (pixels) between HCR-phase, MLR-phase, and PID classifications. Phase partitioning distributions and agreement/mismatch patterns are illustrated in Figure 8 and discussed in Sections 4.3.1–4.3.2. The table of raw sample counts has been moved to the supplementary material, following Reviewer 1's recommendation.

**Please find our response to each reviewer comment in blue.**

**REVIEWER 1 COMMENTS:**

**1. Overview of the paper**

This study analyzes low-level marine boundary layer (MBL) clouds over the Southern Ocean using airborne radar and in-situ data from the SOCRATES campaign. It introduces a method to detect cloud boundaries with radar reflectivity and spectrum width, derives an empirical relationship between reflectivity (Z) and liquid water content (LWC), and classifies cloud phases using radar, temperature, and LWP. Single-layer low clouds (<3 km) made up ~85% of cases. The Z–LWC relation enabled accurate LWP estimates, validated against observations. The cloud phase classification identified 48.8% liquid, 23.3% mixed-phase, and 6.9% ice clouds, and showed good agreement with other established methods. Overall, the study offers a robust radar-based approach for cloud boundary and phase detection, supporting better satellite retrievals and climate modeling over the Southern Ocean.

This resubmitted version of the study presents clear improvements both in writing quality and in the validation of the proposed methodology. The structure is more refined, and the explanations are clearer, which enhances the overall readability of the manuscript. One of the major additions is the comprehensive comparison with multiple existing methods for cloud boundary characterization. This significantly strengthens the robustness of the authors' approach. Based on these elements, I would recommend only minor revisions before final publication.

**2. General suggestions**

The scientific objective is clear, and the methodological approach is generally robust, particularly the development of a radar-based classification for cloud phase. The Supplement is clear and provides important supporting information, especially concerning the in-situ datasets and instrumentation details.

However, several issues hinder the manuscript's clarity and accessibility. First, the text suffers from a lack of consistency in terminology and definitions—key terms and acronyms such as SLW or LOW are introduced without prior explanation. Secondly, there are redundancies and repetitions across the sections, especially in the presentation of the classification results, which could be streamlined for conciseness. Additionally, while the discussion around uncertainties is both important and informative, the writing would benefit from reformulation for clarity and structure. On the methodological side, the rationale for excluding certain cloud types (e.g., fully glaciated clouds) should be more explicitly justified, especially in the context of deriving a Z-LWC relationship. Finally, more attention should be given to harmonizing visual elements (e.g., figure fonts and legends) and providing clearer context in tables and figures, including the comparison of cloud boundaries with in-situ data.

Overall, while the work is of clear interest to the cloud microphysics and remote sensing communities, the revised version appears well-refined, with notable improvements in structure, clarity, and scientific rigor.

We thank the reviewer for providing positive feedback on the manuscript and addressing all their general and specific comments and suggestions, as detailed below, to the best of our ability in the revised version.

As suggested, key terms have been defined before using their acronyms. Repetitions and redundancies have been reduced as far as possible. For the discussion on uncertainties, we have taken into regard the specific comments on each section and make all suggested changes. Figures have been improved for visual and informational clarity. We will also upload each figure file (pdf) separately for the editorial team.

For the choice of cloud profiles, especially for deriving the Z–LWC relationship, we selected only ascending and descending aircraft tracks where the research flight spanned the full vertical extent of the cloud. This ensures that the profiles are representative of the entire cloud column. In-situ measurements from the CDP and 2D-S probes were merged following Zheng et al. (2024) to obtain reflectivity and liquid water content from cloud and drizzle particle size distributions. We did not impose additional constraints beyond the probe specifications, which reliably detect ice particles only above 200 μm. Observed concentrations of particles > 200 μm were very low. Furthermore, below 3 km altitude, no fully glaciated clouds were observed; clouds were predominantly liquid phase or liquid-dominated mixed-phase.

**3. Specific comments and technical corrections**

Abstract.

L.26 : If you are referring to your "LOW" class, you need to define the acronym before using it. Otherwise, you can remove the capital letters or use "low-level clouds".

Corrected as Suggested.

Introduction.

L.35 : This might not be necessary. However, the references are neither in chronological order nor in alphabetical order.

It has been made chronological.

L.38 : "SLW" is not defined before being used.

Corrected as Suggested.

L.53–60 : This paragraph on uncertainties is important and interesting but not very clear. The authors could try to rephrase it.

We have rephrased the paragraph.

L.63 : This is a long sentence (6 lines); please split it into two.

Corrected as Suggested.

L.73 : It's a bit vague—what were these inconsistencies?

We have rephrased the sentence. The machine learning based MLR method to identify cloud phase heterogeneity did not perform accurately in certain cases since machine learning algorithms are not without errors. Further, there were time gaps in the MLR product from time periods where 2D-S measurements were not available. Schima et al., (2022) corrected these *inconsistencies* by a thorough manual investigation of the particle probe imagery data and improved the MLR phase product. Further details can be found in Schima et al., (2022).

Results.

L.131 : Maybe merge this sentence with the one on L.127.

Corrected as Suggested.

L.149 : The ice phase (> 200 µm), although minor, is not used in your method? Is it only useful for comparing your classification with Alessandro's?

No, we do not use the Dp > 200 µm criterion for ice-phase classification in our method. Instead, we primarily rely on a combination of radar and in-situ measurements to determine phase. Similarly, D'Alessandro et al. (2021) and Schima et al. (2022) MLR method derived particle phase from combined in-situ probe measurements rather than using particle size thresholds alone. The mention of the > 200 µm ice-phase classification is only to illustrate the limitation of relying solely on probe-based size thresholds for phase identification and to note that smaller ice particles can also be present within cloud layers.

L.174 : When you refer to a "subset of the dataset", does this correspond to the "5th to 95th percentile interval of the dataset"?

Yes, that is correct. As noted in the preceding lines, we use the 5th to 95th percentile of the dataset together with a kernel density estimation (KDE) method to select the sample density subset used for regression. This approach minimizes the influence of outliers and measurement uncertainties on the estimated Z–LWC relationship.

L.178 : "FIGURE" → "Figure"

Corrected

L.244 : No space between ~ and 22.2%.

Corrected

L.256 : This is essentially what you say in L.226 — consider avoiding the repetition.

Rephrased to avoid repetition

L.265 : You should add the term "mean" as in your table, otherwise it's confusing — it sounds like you're referring to maximum values.

Yes, thanks for noticing this. We have modified this sentence. Note that in the revised version we solely focus on retrieving and evaluating cloud boundaries and LWP only for low-level clouds and all mentions of MID or MOL clouds have been removed in this revision.

L.295 : How do you explain a larger mean difference for cloud top altitude?

In the revised Table 1, the mean differences in $H_{top}$ are much less than previous results. For example, the HCR-derived drizzling $H_{top}$ values are 1.7 km and 1.82 km, respectively, for looking up and down. These $H_{top}$ values are very close to the aircraft in-situ measured $H_{top}$ values (~1.88 km), but slightly higher than the ship-based cloud radar derived $H_{top}$ values (1.5 km). MARCUS cloud-top heights were derived from the ship-based cloud radar observations. Although we focused on all clouds below 3 km between two field campaigns, their derived $H_{top}$ values are compared statistically, not matched spatio-temporally.

 L.314 : "FIGURE" → "Figure"

Corrected

L.319–328 : Formatting issue in the label of Figure 4 and repetition in the text. Please correct this paragraph.

Corrected

L.324 : I understand the importance of stating that you focus on low-level clouds, but this is repeated too many times.

Thanks for pointing this out. We have revised accordingly.

L.324 : Repetition — please revise.

Corrected

L.336 : The authors refer to drizzle in both the text and Figure 5. You mention both "drizzle" and "liquid drizzle" — what is the difference? When you say "drizzle", are you referring to freezing drizzle?

Thanks for noticing that. Although there are some cases of supercooled drizzle, it is indeed redundant to mention drizzle and liquid drizzle separately as drizzle is ideally liquid. We have revised Figure 5 accordingly.

L.330–376 : In this paragraph, the authors mention several thresholds for classifying phases and for Figure 6. Some thresholds are cited with references (Wu et al., 2020a; …) and others are not — how were these chosen?

These thresholds were selected based on multiple existing studies on cloud phase identification, including Shupe (2007), Wu et al. (2020), Xi et al. (2022), Romatschke and Vivekanandan (2022), and Desai et al. (2023), they all are representative of typical MBL cloud macrophysical and microphysical properties. In addition, HCR measurements were examined on a case-by-case basis to refine these thresholds for both drizzling and non-drizzling cloud samples. The final values are consistent with the expected physical ranges for MBL clouds reported in the literature. As noted in the main text, *"To ensure consistency, the phase-diagnostic thresholds adopted in this study were compared with values reported in previous studies (e.g., Shupe, 2007; Xi et al., 2022; Romatschke & Vivekanandan, 2022; Desai et al., 2023)."* These supporting studies have been cited in the manuscript where relevant.

L.377 : Does this kind of "misclassification" occur frequently? Can the authors quantify this? Does a cloud with LWP $\geq$ 20 g/m² and LWCs > 0.2 g/m³ necessarily exclude the presence of ice?

This is an important question. We can quantify this by comparing the liquid phase identified in our method with the PID method. In our dataset, the classified liquid phase shows very low coincidence with the PID-classified frozen phase (~2.2%), indicating that such "misclassification" occurs only rarely. Large LWC (> 0.2 g/m³) and LWP ($\geq$ 20 g/m²) values are generally consistent with the liquid phase, following Shupe (2007), where the liquid layer is defined over a depth yielding an average LWC of 0.2 g/m³. This step is crucial to accurately identifying liquid phase and ideally excludes most ice-phase particles, which exhibit lower LWC and LWP values. In clouds with high LWC and LWP, the precipitation phase is typically drizzle or rain (occasionally supercooled or freezing rain). Analyzing the probe-based cloud particle phase classifiers (from MLR-method, and UWILD classification by Atlas et al., 2021), we find that liquid-phase cloud droplets demonstrate high LWC/TWC fractions (> 0.9), whereas ice-phase clouds have high IWC/TWC fractions (> 0.9), consistent with the findings of Korelov and Milbrandt (2022). While the presence of ice in such conditions is not impossible, our analysis suggests it is rare.

L.448 : The authors could add a sentence at the beginning of paragraph 4 stating that only low-level clouds are used, in order to avoid repeated reminders later.

Noted. We have removed the repeated mentions to low-level clouds in section 4.

L.481 : The concept of scale plays a crucial role in method comparison — the authors should emphasize this point more clearly in the text.

We have revised the comparison of in-situ probe-based phase estimation (MLR) with the HCR-phase (developed in our study) entirely. These scale differences affect the comparison in three main ways: 1) The in-situ probes do not provide phase information across the full vertical extent of HCR-observed cloud profiles, 2) the spatial separation between the in-situ probes and the radar observations, 3) the limited number of overlapping samples between the two methods (within a ~200-meter window). This has been highlighted consistently in Section 4.3.1, and other parts of the main text.

L.525 : The authors should describe Table 2 in more detail. Otherwise, it may not be necessary to keep it in the main text; consider moving it to the supplementary materials, since you only mention it once.

This is a good observation. The raw sample counts in the original Table 2 do not add any extra information beyond what is already discussed in Sections 4.3.1 and 4.3.2, along with Fig. 8. We have revised Table 2 accordingly to retain only the hit rates (agreement percentages) for overlapping samples between the HCR-phase and PID scheme, and the nearest comparable samples for the HCR-phase and MLR method. The comparisons of the raw sample counts for overlapping samples (or pixels) have been moved into the supplementary.

L.574 : The percentages are calculated on classes that are not present in both methods (Mix % and Melting %), which could explain the larger differences.

Yes, that is correct. We have added this line to the main text.

L.582 : Exactly — you could perhaps add that the mixed-phase strongly depends on the observation scale (microphysical <--> macrophysical).

We have rephrased the sentence as *"Mixed-phase cloud identification remains the most uncertain, particularly in regions with high spatial heterogeneity, and is strongly influenced by the observation scale (from microphysical to macrophysical)."*

Figures.

Figure 3 : The font size for y-ticks is not the same in panels f and h. You could also increase the x-ticks size in panel h to match the one in subplot d.

Corrected this in revised Figure 3.

Figure 4 : There is an issue in the label: "4 Low-level cloud phase classification method and discussion".

Corrected.

**REVIEWER 2 COMMENTS:**

The authors use in-situ observations and remote sensors to classify hydrometeors observed during the SOCRATES field campaign over the Southern Ocean. While the objective of the study is of high interest, unfortunately, it is not very well carried out, as outlined in the General Comments below. The proposed algorithms are based on interesting ideas but the results are not convincing. A much more rigorous evaluation of the proposed algorithms is needed before possible publication. Also, the authors do not sufficiently explain why their method is superior to previously published methods, i.e., why this new method is needed.

We thank the reviewer for their valuable scrutiny of our paper and have extensively revised the manuscript, addressing their comments and suggestions to the fullest extent possible within the scope of this study.
    Although all classification methods (including ours) have inherent uncertainties and limitations, the strength of our approach lies in combining in-situ and radar measurements to identify cloud boundaries and phase in a straightforward manner based on airborne observations. This addresses vertical coverage and attenuation issues that can affect other observational approaches—for example, lidar signals can experience strong attenuation in cloud layers, particularly in nadir-viewing configurations, and in-situ probes cannot capture the full vertical extent of cloud profiles—information that radar observations can provide. The need for our method arises from addressing limitations in existing phase-identification techniques for MBL clouds over Southern Ocean, which often rely on a single observational platform and can be affected by attenuation under certain conditions. Leveraging the SOCRATES aircraft dataset, our approach integrates airborne radar and in-situ observations to produce vertically continuous phase classifications that remain robust in the optically thick, low-level clouds over the SO and other regions.

**General Comments:**

1. The authors claim in the conclusion that "Both the cloud base detection and phase classification methodologies were rigorously evaluated against existing methods.". Unfortunately, I have to disagree with these statements. For the most part, the authors only compare bulk statistics of cloud base height or particle phase, which is not particularly meaningful. A bulk comparison does not say much about the accuracy for individual cases. It only compares averages, and even those averages are skewed, given that the radar-derived quantities are time-height cross sections while the in-situ probes measure only along the flight path. The only rigorous evaluation is the pixel-by-pixel evaluation against the PID method - which reveals significant differences. Without a pixel-by-pixel comparison with the cloud probe-based methods, the value of the algorithms cannot be confirmed.

We acknowledge the reviewer's concern and agree that a pixel-by-pixel comparison is the most rigorous means of evaluating phase classification performance. The pixel-by-pixel comparison with the PID scheme was conducted because it was the only dataset with overlapping samples available. Differences between the two methods are expected due to their distinct classification methodologies The PID scheme applies a fuzzy-logic hydrometeor classification using multi-sensor inputs. Some of these output classes are broad categories such as cloud, precipitation, and melting, which can be somewhat ambiguous for certain applications. In contrast, the HCR-phase

(developed in our study) algorithm is a threshold-based, radar-only phase classifier constrained by LWC/LWP and temperature and can further refine these broader PID categories into specific phase types (liquid, mixed, ice, drizzle, rain, snow). For the pixel-by-pixel evaluation, we mapped both outputs to the common phase classes (liquid, frozen, drizzle, rain) to ensure comparability. An ambiguity arises from the PID "melting" and HCR "mixed-phase" classes, which are unique to their respective classification methods and absent in the other, and could explain some of the larger differences in phase sample percentages. Differences between the two methods therefore reflect both their underlying decision logic and the type of input data used.

For MARCUS, comparisons were carried out over a spatiotemporally collocated Southern Ocean region with uniform spatial resolution. Although these were not exact track overlaps, they still provide a reliable basis for inferring dominant cloud microphysical properties.

Following the reviewer's suggestion, we have now included a pixel-by-pixel comparison with the cloud probe–based MLR method in the revised manuscript (Section 4.3.1, Table 2a, Table S3, Figure 8a). HCR-phase retrievals are performed only on fully sampled cloud profiles observed from HCR. MLR-phase data from the CDP and 2D-S probes within 100–200 m of the first valid HCR range gates were extracted, and the corresponding HCR-phase data were grouped into three categories (liquid, mixed, ice). The sharp sawtooth tracks where the in-situ probes sampled the full vertical extent of the cloud were excluded in this study because these profiles can not match with HCR observed profiles (HCR could not observe cloud profiles when aircraft ascended or descended). HCR can observe complete cloud profiles when aircraft flew either above (nadir view) or below (zenith view) the cloud layer. Although we could compare in-situ cloud base and top heights from sawtooth tracks with the nearest HCR-derived cloud boundaries (as they represent the same cloud profile), such a direct comparison is not feasible for HCR-phase, which requires collocated phase samples within ~200 m to ensure meaningful comparisons.

In this revision, there are a total of 298 valid samples selected for the MLR–HCR comparison, corresponding to flight segments passing very close to or along cloud edges. Two main factors limit the direct comparison with probe-based analysis: (1) spatial separation between in-situ probes and radar observations (Romatschke and Vivekanandan, 2022) and (2) the small number of overlapping samples which can potentially lead to statistical variabilities. HCR's fixed nadir- or zenith-pointing configuration further reduces overlap during aircraft ascents or descents.

2. The cloud-base detection algorithm is strongly based on spectrum width, which is a very noisy quantity. I find it unlikely that it would actually give the desired results and unless the authors provide a convincing rigorous evaluation showing the opposite, I would be very surprised if it were possible to derive cloud base from HCR spectrum width.

Thanks for raising this important point. While the spectrum width (WID) gradient–based method performed well for most drizzling and non-drizzling cases, we did observe some instances where it did not work as accurately, particularly during strong precipitation events (such as rain). In some of those cases, the WID-gradient method identified drizzle base instead of true cloud base. As the reviewer correctly noted, this was a limitation that needed to be addressed. In response, we have revisited and completely revised the cloud-base detection approach. The updated method used to determine cloud base is based on a combination of HCR reflectivity (dBZ), Doppler velocity (Vd), and WID measurements for each vertical cloud profile on a case-by-case basis. This method is specifically tuned for four scenarios: drizzling and non-drizzling clouds, each for upward- and downward-looking radar/lidar configurations.

The updated cloud-base detection method can be summarized as:

1. For non-drizzling cases (i.e., all reflectivity values in a column < –15 dBZ), $H_{base}$ is defined as the lowest altitude with a valid radar reflectivity return.

2. In drizzling cloudy cases (e.g., virga), $H_{base}$ is estimated as the altitude where dBZ and Vd (and/or WID) show a marked increase. A threshold of Vd or WID > 0.5 m s$^{-1}$, combined with a local maximum gradient in dBZ, is used to identify $H_{base}$, marking the transition from a cloud layer dominated by small droplets to a drizzle layer with larger falling particles. This is followed from the reasoning that, above the cloud base, reflectivity contains contributions from both cloud droplets and drizzle drops, whereas below it, it primarily reflects drizzle or precipitation. Near the cloud base, droplets may grow into drizzle through collision–coalescence, producing a sudden increase in reflectivity and Vd (or WID) due to the presence of larger particles and stronger downdrafts. Elevated WID or Vd indicates greater turbulence and a wider range of particle velocities, consistent with drizzle or precipitation. Below the base, drizzle drops may be partially or fully evaporated in subsaturated conditions, causing a decrease in reflectivity with decreasing altitude (similar to the profile shown in Fig. 2b of Wu et al., 2020). For special case arises when dBZ exceeds 5 near the surface, indicating strong precipitation (e.g., rain; Shupe, 2007). In such profiles, the true $H_{base}$ is determined by examining the vertical gradient of dBZ together with Vd (and/or WID) values above the first altitude where dBZ exceeds 5 dBZ.

The Figures below show the cloud base for selected cases of non-drizzling and drizzling cloud profiles.

[Figure]

3. For the phase classification, the authors rely heavily on the derived LWC-Z relation. I do not find the arguments that such a relation should exist very convincing. If I understand the method correctly, it attributes all variations in Z along the radar beam to variations in LWC. What is the basis for this assumption? Again, I could be convinced otherwise, but only with a better evaluation of the algorithm.

The basis for the LWC–Z relationship in our method comes directly from in-situ probe measurements collected during the SOCRATES campaign. Specifically, we analyzed CDP and 2D-S DSD spectra at 1 Hz from ascending and descending aircraft tracks where the aircraft sampled the full vertical extent of the cloud layers. From these profiles, we derived in-situ Z, LWC, and LWP, focusing on uniform, single-layer low-level stratocumulus MBL clouds. We based our analysis on multiple existing studies that derived empirical Z–LWC relationships from in-situ measurements (e.g., Dong & Mace, 2003; Mioche et al., 2017; Oh et al., 2018; Vivekanandan et al., 2020). Following this approach, we used our collocated probe measurements to establish an empirical Z–LWC relationship for the SOCRATES dataset, which was then applied to HCR radar reflectivity measurements to estimate LWC along the beam at each range gate. This technique is

particularly valuable in the absence of radiometer-based measurements, as it enables the derivation of time–height profiles of LWC and the estimation of column-integrated LWP across the cloud vertical column. The underlying assumption, that radar reflectivity and LWC are direct functions of liquid cloud particle size distribution ($\propto D$), and therefore intrinsically linked through the microphysical properties of the cloud, has been well established in the cited literature.

Furthermore, the in-situ Z and LWC measurements were constrained to only $5^{th}$ to $95^{th}$ percentile of the dataset to minimize the influence of extreme outliers. A kernel density estimation (KDE) was used to estimate relative point density in the Z-LWC (log) space. Due to the noisy nature of the dataset where larger particle diameters return extremely high Z values ($\sim D^6$) and relatively lower LWC values ($\sim D^3$), a log-log linear regression was performed only using a subset of the dataset with high sample density to minimize measurement uncertainties. The predicted LWCs from the LWC–Z relationship agrees well with in-situ measured LWC (means: 0.21 vs. 0.27 g/m³; RMSE ≈ 0.03 g/m³), and the corresponding predicted LWPs (means: 93.73 vs. 95.55 g/m²) have an RMSE of ~12 g/m².

4. To make this study publishable, the authors would need to do the following:

a) Evaluate the cloud-base algorithm on a ray-by-ray basis under precipitating conditions against, e.g., the GV-HSRL. If the results don't agree, the algorithm is likely flawed and should not be used.

Following the reviewer's suggestion, we have revised cloud-base detection thoroughly. The new cloud bases have been evaluated on a ray-by-ray basis under drizzling (precipitating) and non-precipitating conditions by comparing with the HSRL-derived cloud bases, the MARCUS MPL/Ceilometer-derived cloud bases, and the in-situ cloud bases estimated from the nearest adjacent sawtooth tracks of the aircraft. The findings have been presented in Figure 3 and Table 1 (Section 3.2) of the revised manuscript.

Here are brief summaries of updated results from the revision. As illustrated in Figure 3 for the selected cases and statistical results listed in Table 1, HSRL- and HCR-derived $H_{base}$ heights show good agreement, with HSRL values generally slightly higher than HCR for both drizzling and non-drizzling for all looking-up cases (aircraft flew below cloud layer). The mean differences between HCR- and HSRL-$H_{base}$ are 0.25 km and 0.21 km, respectively, for drizzling and non-drizzling cloud cases. These offsets arise because the HSRL backscatter signal often exhibits its steepest slope a few range gates above the HCR-derived base, leading to higher HSRL estimates. The differences reflect the instruments' sensitivities: lidar responds to the second moment of the particle size distribution, whereas radar responds to the sixth moment. Consequently, radar-derived bases are often lower than lidar-derived ones, especially for drizzling clouds (Dong et al., 2005). For non-drizzling clouds, lidar may miss thin cloud edges near the true base due to low particle concentrations or signal-to-noise limitations, while radar can detect hydrometeors near the base, including evaporating particles or low concentrations of small droplets.

As for looking-down cases, the HSRL signals are often attenuated rapidly below the cloud top and cannot penetrate deeper into the cloud layer, preventing reliable detection of cloud base in these cases (~80% of the total aircraft samples). This is visualized in Figures 3c, 3f, and S2 (supplementary) through the HSRL-backscatter plots. Therefore, for looking-down cases, the HCR- and HSRL- $H_{base}$ heights exhibit significant differences. Note that we have tweak the HSRL-backscatter thresholds differently for drizzling and non-drizzling cases by studying the slopes of

the backscatter signal for each case in this revision following the studies of Wang and Sassen (2001), Shupe (2007), Flynn et al. (2020), and Kang et al. (2024).

[Figure]

Performing a regression analysis on the 20–80th percentile of the HCR- and HSRL-$H_{bases}$ with KDE density approximation to retain the densest data sample (to reduce the influence of extreme outliers and focus on the central distribution), gives a correlation of 0.74 (RMSE 0.15 km) for drizzling, and 0.70 (RMSE 0.25 km) for non-drizzling looking-up cases.

To further evaluate the HCR-derived cloud boundaries, the in-situ measured cloud bases and tops from SOCRATES sawtooth flight segments were used where the aircraft sampled full cloud profiles (Kang et al., 2024). The comparisons used in this study are limited to the in-situ boundaries located nearest to the HCR-derived $H_{base}$ and $H_{top}$ for complete, non-intersected profiles. The reasoning is that the cloud profiles span a broader time axis, so the nearest sawtooth segment still samples the same stratocumulus cloud, with comparable cloud base heights. Valid in-situ cloud samples were defined by cloud-droplet number concentration ($N_c$) > 5 cm$^{-3}$ and $LWC_c$ > 0.01 g m$^{-3}$ (Wood, 2005; Zheng et al., 2022). Precipitating/drizzling cases were identified when valid drizzle samples (drizzle-drop number concentration $N_d$ > 0.001 cm$^{-3}$) occurred below the cloud base (Zheng et al., 2024). Twenty-nine such cases were selected (Table S2). An example is shown in the figure below. As shown in Table 1, The mean differences of $H_{base}$ between HCR and in-situ are ~0.07 km for both drizzling and non-drizzling cases. These demonstrated level of agreements further prove that the cloud-base detection algorithm from HCR measurements is robust and reliable. Some differences arise because sawtooth flight paths span the full vertical extent of cloud layers, while HCR detection shows a ~100 m offset relative to in-situ probes. **Note that such comparison, however, is not possible for probe-based phase classification with HCR-phase method.**

[Figure]

b) Evaluate the phase classification on a pixel-by-pixel basis (near the aircraft in in-cloud flight legs) against the particle probe derived results from previous studies. The evaluation against PID is good and necessary but only an evaluation against a different sensor will show the true potential of the method.

Please refer to the response to General Comment 1. Following the reviewer suggestion, we have performed a pixel-by-pixel comparison of HCR-phase with the probe-based phase classifier i.e. the MLR-phase product. As discussed, such a comparison is not straightforward for our selected cloud profiles for estimating phase, which are cloud profiles where the HCR can view the cloud layer completely (either above or below cloud layer). Therefore, the comparison could only be performed over 298 selected samples, corresponding to flight segments passing very close to or along cloud edges (with HCR-classified phase). Furthermore, for enabling the comparisons, the corresponding HCR-phase data within this offset from the probe measurements were manually adjusted using a homogenizing box filter to identify the dominant phase across the first 2–3 nearest radar range gates, and then classified into three broad categories: liquid, mixed, and ice. For instance, drizzle and rain were categorized as liquid, while snow was reclassified as ice. Across these nearest overlapping samples, both methods demonstrate a reasonable level of agreement with hit rates of 59% for liquid phase, 71% for mixed phase, and 48% for ice phase. Overall, both methods demonstrate a combined hit rate of ~60% across all phase samples. The hit rates (or sample agreement %) for each comparable phase category have been calculated and presented in Table 2a, while specific sample count comparisons for each phase category between the HCR-phase classification and the MLR-method are summarized in Table S3 (Supplementary). The comparison between the phase-samples from the two methods have also been illustrated in the bar plot in Fig 8a. The figure below illustrates a specific case of comparison between HCR-phase and probe-based MLR phase.

[Figure]

Fig. The contour plot is from HCR-phase (tuned for comparison) while the dots are the probe-classified phase (from MLR-phase product). The color schemes are: Red (Ice), Green (Mixed), Blue (Liquid)

**Given the inherent limitations and uncertainties that affect direct comparisons with the probe-based MLR-phase product, the HCR-phase vs. PID-scheme evaluation remains the only pixel-by-pixel comparison that can be conducted with high reliability.** This approach ensures consistent sampling, minimizes mismatches due to spatial and temporal offsets, and directly tests the classification merits of the HCR-phase method. As such, it provides a robust basis for assessing the strengths and weaknesses of the HCR-phase retrieval, and additional comparisons with less compatible datasets would add uncertainty without offering meaningful new insight.

**Specific comments:**

5. Title: It is not ideal to use an acronym in the title (MBL).

Corrected the title as suggested.

6. Line 23: Why is LOW capitalized?

Changed all LOW to low-level or low clouds.

7. Line 38: Define SLW.

Corrected

8. Line 55: The statement "onboard radar and lidar experience less attenuation than ground-based sensors" is a bit misleading and should be clarified.

Rephrased this to 'Airborne radar and lidar also experienced less signal attenuation than their ground-based or space-borne counterparts (Dong et al., 2025; Ewald et al., 2021), due to their measurement paths typically passing through a shorter atmospheric column, reducing the cumulative effects of absorption and scattering by hydrometeors, atmospheric gases, and surface clutter.'

9. Line 62: MBL has already been defined.

Corrected

10. Line 65 (and later): These instruments are generally known as HCR (not GV-HCR) and GV-HSRL (not NCAR-HSRL). Please be consistent throughout the paper.

Thanks for pointing this out. Corrected as suggested.

11. Line 81: At W-band and at the high vertical resolution of HCR, surface clutter is insignificant.

We have rephrased this sentence to make it clearer.

12. Line 97: It is not clearly stated why yet another cloud classification algorithm is needed. What are the expected improvements over the existing ones?

Thanks for pointing this out. The line has been rephrased accordingly as per suggestion.
        We acknowledge the existence of various cloud phase classification methods in our manuscript, each tuned to specific datasets, platforms, and regions, which naturally produce different results. However, phase identification for low-level SO clouds remain limited, and further development of robust, broadly applicable methods is needed.
Our approach is motivated by two factors: (1) developing a straightforward classification method that combines radar and in-situ measurements, offering a simpler approach that overcomes the limitations of standalone probe-based classifications and remains effective even where lidar suffers from signal attenuation, and (2) leveraging the SOCRATES aircraft dataset, which provides near-direct cloud sampling and parameters less affected by attenuation. This work is an investigative

step toward addressing persistent challenges in determining cloud phase and macrophysical properties over the SO, complementing rather than replacing existing approaches.

13. Line 127: The temporal resolution of HCR and the GV-HSRL is 2 Hz in the combined dataset.

Yes, but we resample it to 1 Hz for our study.

14. Line 131: This statement is a bit misleading. The GV-HSRL actually detects more clouds, especially thin clouds, because of sensitivity limitations of HCR. However, as you state, it is highly attenuated.

It is correct that the GV-HSRL detects more thin clouds due to its higher sensitivity. However, much of the HSRL signal is concentrated near cloud edges and is strongly attenuated, so it does not penetrate the full cloud depth. In contrast, HCR samples the majority of the vertical extent of the cloud profile. Consistent with this, we find only ~11% radar–lidar overlap in our dataset.
    We have added this sentence 'The HSRL has an advantage over HCR in detecting thin cloud layers, but it attenuates strongly near cloud edges depending on viewing direction and cannot penetrate the full vertical extent of the cloud layer, which the HCR can.'

15. Line 133: 3D -> 2D.

Corrected

16. Section 2.2: One major issue with the Z-LWC relation is attenuation of the radar signal, which is significant at W-band. How does this method deal with attenuation?

It is correct that the W-band experiences some attenuation for deep convective clouds. However, the Z–LWC derivation in this study is based primarily on in-situ probe measurements of MBL clouds. For the HCR data, we used the quality-controlled valid-cloud mask as described in Romatschke (2021), which retains only valid cloud echoes and removes noise and highly attenuated signals. As Romatschke (2021) notes, "when HCR observes thick, high-liquid-water-content clouds (e.g., in convection), the signal may be fully attenuated before penetrating the full cloud depth. In nadir mode, the algorithm searches for an ocean or land surface echo; if this is below a set threshold or absent, the region from the cloud base (last valid echo) to the radar range limit is flagged as extinct, indicating a high likelihood—but not certainty—of cloud or precipitation in that region". These flagged points are excluded in our analysis, and the HCR–HSRL Moments dataset we used already incorporates this filtering.
    Following the reviewer's suggestion, we tested our classification method on a small sample of attenuation-corrected HCR profiles (see figure below) from SOCRATES provided by Gwo-J Huang (NSF NCAR, EOL). Because our classification is threshold-based and uses quality-controlled HCR data that already excluded highly attenuated signals, we found no significant differences (<2%) in classification results, and derived LWC, LWP for this sample. The HCR-dBZ difference being < 5dB. Note the figure shown below are raw dBZ profiles (corrected for attenuation) and not quality controlled to retain only valid cloud flags.

[Figure]

The figure below is taken from Romatschke (2021) with the FLAG fields defined for a sample from **SOCRATES, and the corrected DBZ_MASKED profile.** That being said, correcting for liquid attenuation is not very straightforward and requires estimates of atmospheric variables (like temperature and humidity) which are not measured directly by the radar. **Therefore, we do not apply additional attenuation correction, and it is beyond the current scope of this study.**

[Figure]

17. Section 3.3: I am not convinced by the identification method for Hbase. The difficulty is the distinction between cloud echoes and echoes from precipitation shafts underneath the clouds. The histograms comparing with MARCUS observations do not sufficiently show that the method is working because they do not distinguish between precipitating and non-precipitating clouds. The fact, that GV-HSRL derived Hbase values are significantly higher than the HCR derived values likely shows that the HCR-based algorithm mis-identifies precipitation shafts as clouds. It would be helpful to focus only on radar beams with surface precipitation and carefully compare those with HSRL and MARCUS observations. Some plots demonstrating the method under precipitation conditions would also be helpful (Figure 3 is not detailed enough to provide insight into the quality of the algorithm). I am very doubtful that a method base on spectrum width can work because many different phenomena contribute to the observed spectrum width values and it is very difficult to tease the different contributions apart. Overall, I am wondering if the identification of Hbase is even needed for the current study? Given all the question marks regarding the method, I am wondering if it would be better to exclude it from the manuscript.

The cloud boundary identification method enables determination of cloud top and base heights from HCR measurements in the absence of radiosonde or dropsonde data. While lidar-based cloud base detection is only valid in zenith-pointing mode, the radar-only approach provides reliable, simultaneous detection of both boundaries regardless of instrument orientation. Furthermore, it also provides the basis for LWP estimation, required for phase classification.

However, we acknowledge that the spectrum width–based method showed inconsistencies. This led us to fully revise the cloud-base detection approach, using HCR reflectivity, Doppler velocity, and spectrum width profiles, and evaluating them case-by-case for both precipitating and non-precipitating clouds. Furthermore, the cloud boundary heights estimated from the new method has been thoroughly evaluated through comparisons with HSRL-, MARCUS- and in situ- derived cloud boundaries, for precipitating and non-precipitating cases (and illustrated in new Fig 3 of the main text).

**Please refer to the response to General Comments 2, and 4a, for the detailed explanation.**

18. Line 326: How are "noisy pixels" defined and identified?

In this study, "noisy pixels" refer to thin, isolated cloud detections that are not physically connected to the main cloud structure, or appear as narrow filaments attached to it, which can introduce inconsistencies in boundary detection. These are identified by assessing the spatial continuity of cloud pixels within each HCR profile. Profiles containing such returns are excluded, and only fully sampled HCR profiles are retained. Additionally, sharp sawtooth flight segments are removed, as HCR does not reliably capture cloud boundaries in those cases. Profiles with cloud tops exceeding 3 km are also excluded. For analysis, we retain only single-layered cloud profiles, ideally representing the first observed cloud layer from the surface.

[Figure]

19. Figure 7: Please list the exact date and time of this example. RF 13 took place on February 20th but I could not find this example at the given time.

Yes, it was typo mistake, thanks for pointing this out. The figure was from RF12 (Feb 18, 2018) not RF13. We have updated the Figure 7 with a sample from RF7 (Jan 31, 2018). Note that the exact timestamps might shift slightly from the 1 Hz measurements as we used a 30 second averaging for our analysis.

20. Sections 4.3.1 and 4.3.2: I am not sure what this comparison is supposed to tell us. Comparing bulk fractions of different phases is not meaningful given the differences in sampling (1D vs. 2D). For a true comparison, the authors would need to compare actual data points from in-cloud flight legs with HCR observations in close proximity. Then they could calculate some kind of hit/miss table or similar.

We acknowledge the reviewer's concern. However, even with certain limitations, the bulk fraction comparison still provides valuable insight into the general phase partitioning and associated micro- and macrophysical characteristics of low-level clouds over the SO. In particular, MBL clouds in this region consistently show a dominant contribution from liquid clouds (often supercooled liquid water), followed by mixed-phase clouds containing both liquid and ice particles within the same cloud volume. This behavior is captured consistently across all the compared phase-classification methods. In the revised manuscript, comparison with bulk fractions of differences is primarily conducted for the MARCUS-based phase products, where exact overlapping samples required for a pixel-by-pixel evaluation are not available. In these cases, the bulk statistical comparison provides a practical means to characterize overall phase distributions and their relative contributions, complementing the collocated comparisons presented elsewhere in this study.

**Further, please refer to our responses to General Comments 1 and 4b.** The phase classification in this study is restricted to fully sampled cloud profiles—i.e., cases where the HCR captures the full vertical extent of the cloud. This constraint, along with the exclusion of sawtooth sections (which overlap strongly with the probe sampling altitude), makes direct comparison with the probe-based MLR-phase product challenging. As a result, only 298 valid samples were available for comparison, corresponding to flight segments passing very close to or along cloud edges. Given these inherent limitations and uncertainties, the HCR-phase vs. PID-scheme evaluation remains the only pixel-by-pixel comparison that can be conducted with high reliability.

Following the reviewer's suggestion, we computed hit rates following Romatschke and Vivekanandan (2022), with results presented in Table 2. Phase-category-specific comparisons are shown in Fig. 8, and exact phase sample (pixel) count comparisons are provided in Tables S3 and S4, for HCR-phase vs. the probe-based MLR-phase and PID scheme. Miss rates can be estimated as 100 – hit rate (%).

For the nearest comparable probe-based MLR-phase samples (within 200 m offset), HCR-phase achieved hit rates of 59 % for liquid phase, 71 % for mixed phase, and 48 % for ice phase, with an overall agreement of ~60 % across all phase samples. Comparisons between HCR-phase and PID show substantial agreement over the overlapping pixels, with hit rates of 75 % for liquid, 52 % for frozen, 66 % for drizzle, and ~100 % for rain. The combined hit rate across all the comparable categories is ~70%.

21. Section 4.3.3: The comparison with the PID method is interesting but shows significant differences. Only a pixel-by-pixel comparison with a different instrument (as was done for the PID method) will reveal which method actually works better.

Following the reviewer's suggestion, we performed a direct comparison between the HCR-phase classification and the probe-based MLR-phase product for spatially comparable samples, defined as probe measurements within 100–200 m of the nearest HCR range gate. Differences in phase partitioning, even for overlapping or nearest-overlapping samples, are expected and likely arise from inherent differences in underlying classification methodology, decision logic, and observational scale. These differences are however within acceptable margins and reflect the expected variability between the independent methods, highlighting a greater margin of agreement within measurement uncertainties. Please also refer to the detailed responses to the other relevant comments.

A key challenge in comparing in-situ probe-based phase classifications with the HCR-phase method lies in differences in observational scale and sampling coverage. The limited number of spatially comparable samples increases statistical variability. The MLR classifiers are trained on microphysical probe data collected at the GV aircraft's altitude and therefore cannot characterize the full vertical extent of HCR-observed cloud profiles, especially when the aircraft is above or below the cloud. Moreover, the MLR algorithm is designed for subfreezing conditions; samples at temperatures above 0 °C (though rare) remain unclassified and are excluded from final phase statistics, further limiting one-to-one comparisons. Despite these constraints, the comparison offers a useful approximation of how in-situ measured cloud phases relate to HCR-derived phases. Additionally, for HCR-phase vs. PID, a major difference arises in cases with supercooled liquid droplets coexisting with ice particles. HCR-phase often classifies such conditions as mixed-phase based on low reflectivity, low Doppler velocity, and high LWP, whereas PID does not include any LWP constraints. Additionally, HCR-phase explicitly categorizes drizzle, rain, and snow below the cloud base for precipitating cases, while PID applies no cloud-boundary constraints in its classification.

22. Line 584: The authors claim that the HCR-phase method has strong capabilities for detecting mixed-phase clouds. However, this has not been shown or verified. Just because the method classifies many pixels as "mixed" does not mean that this represents reality.

The significant presence of mixed-phase clouds over the Southern Ocean has been well-documented in previous studies, including Xi et al. (2022) and Mace et al. (2021). In this study, mixed-phase clouds occur most frequently between −15 °C and −2.5 °C, underscoring the spatial heterogeneity of low-level stratocumulus. This is consistent with the findings of D'Alessandro et al. (2021) and Maciel et al. (2024). The statement in question refers to a specific advantage of the HCR-phase method: when both ice and liquid hydrometeors coexist within the same cloud volume, incorporating an LWP constraint allows for more robust discrimination of mixed-phase conditions from purely ice or liquid cases. On the other hand, the PID and MLR methods do not incorporate any LWP-based criterion.

**REFERENCES**

Atlas, R., Mohrmann, J., Finlon, J., Lu, J., Hsiao, I., Wood, R., & Diao, M.: The University of Washington Ice-Liquid Discriminator (UWILD) improves single-particle phase classifications of hydrometeors within Southern Ocean clouds using machine learning, Atmospheric Measurement Techniques, 14(11), 7079–7101, https://doi.org/10.5194/amt-14-7079-2021, 2021.

D'Alessandro, J. J., McFarquhar, G. M., Wu, W., Stith, J. L., Jensen, J. B., & Rauber, R. M.: Characterizing the Occurrence and Spatial Heterogeneity of Liquid, Ice, and Mixed Phase Low-Level Clouds Over the Southern Ocean Using in Situ Observations Acquired During SOCRATES, Journal of Geophysical Research: Atmospheres, 126(11), https://doi.org/10.1029/2020JD034482, 2021.

Desai, N., Diao, M., Shi, Y., Liu, X., & Silber, I.: Ship-Based Observations and Climate Model Simulations of Cloud Phase Over the Southern Ocean, Journal of Geophysical Research: Atmospheres, 128(11), https://doi.org/10.1029/2023jd038581, 2023.

Dong, X., & Mace, G. G.: Profiles of Low-Level Stratus Cloud Microphysics Deduced from Ground-Based Measurements, Journal of Atmospheric & Oceanic Technology, 20(1), 42-53, https://doi.org/10.1175/1520-0426(2003)020<0042:POLLSC>2.0.CO;2, 2003.

Dong, X., Minnis, P., and Xi, B.: A climatology of midlatitude continental clouds from the ARM SGP Central Facility: Part I: Low-level cloud macrophysical, microphysical, and radiative properties, J. Climate, 18, 1391–1410, https://doi.org/10.1175/JCLI3342.1, 2005.

Dong, X., Das, A., Xi, B., Zheng, X., Behrangi, A., Marcovecchio, A. R., & Girone, D. J.: Quantifying the differences in Southern Ocean clouds observed by radar and lidar from three platforms. Geophysical Research Letters, 52, e2024GL112079. https://doi.org/10.1029/2024GL112079, 2025

Ewald, F., Groß, S., Wirth, M., Delanoë, J., Fox, S., & Mayer, B.: Why we need radar, lidar, and solar radiance observations to constrain ice cloud microphysics, Atmospheric Measurement Techniques, 14(7), 5029–5047, https://doi.org/10.5194/amt-14-5029-2021, 2021.

Flynn, D., Sivaraman, C., Comstock, J., and Zhang, D.: Micropulse Lidar Cloud Mask (MPLCMASK) Value-Added Product for the Fast-Switching Polarized Micropulse Lidar, Technical Report, https://doi.org/10.2172/1019283, 2020.

Kang, L., Marchand, R. T., & Wood, R.: Stratocumulus precipitation properties over the Southern Ocean observed from aircraft during the SOCRATES campaign, Journal of Geophysical Research: Atmospheres, 129(6), e2023JD039831, https://doi.org/10.1029/2023JD039831, 2024.

Mace, G. G., Protat, A., and Benson, S.: Mixed-phase clouds over the Southern Ocean as observed from satellite and surface-based lidar and radar, J. Geophys. Res.-Atmos., 126, e2021JD034569, https://doi.org/10.1029/2021JD034569, 2021.

Maciel, F. V., Diao, M., and Yang, C. A.: Partition between supercooled liquid droplets and ice crystals in mixed-phase clouds based on airborne in situ observations, Atmos. Meas. Tech., 17, 4843–4861, https://doi.org/10.5194/amt-17-4843-2024, 2024.

Mioche, G., Jourdan, O., Delanoë, J., Gourbeyre, C., Febvre, G., Dupuy, R., Monier, M., Szczap, F., Schwarzenboeck, A., and Gayet, J.-F.: Vertical distribution of microphysical properties of Arctic springtime low-level mixed-phase clouds over the Greenland and Norwegian

seas, Atmos. Chem. Phys., 17, 12845–12869, https://doi.org/10.5194/acp-17-12845-2017, 2017.

Oh, S.-B., Lee, Y. H., Jeong, J.-H., Kim, Y.-H., and Joo, S.: Estimation of the liquid water content and Z–LWC relationship using Ka-band cloud radar and a microwave radiometer, Meteorological Applications, 25, 423–434, https://doi.org/10.1002/met.1710, 2018.

Romatschke, U., Dixon, M., Tsai, P., Loew, E., Vivekanandan, J., Emmett, J., and Rilling, R.: The NCAR Airborne 94-GHz Cloud Radar: Calibration and Data Processing, Data, 6, 66, https://doi.org/10.3390/data6060066, 2021.

Romatschke, U., and Vivekanandan, J.: Cloud and precipitation particle identification using cloud radar and lidar measurements: Retrieval technique and validation, Earth and Space Science, 9, e2022EA002299, https://doi.org/10.1029/2022EA002299, 2022.

Schima, J., McFarquhar, G., Romatschke, U., Vivekanandan, J., D'Alessandro, J., Haggerty, J., et al.: Characterization of Southern Ocean Boundary Layer Clouds using airborne radar, lidar, and in situ cloud data: Results from SOCRATES, Journal of Geophysical Research: Atmospheres, 127, e2022JD037277, https://doi.org/10.1029/2022JD037277, 2022.

Shupe, M. D.: A ground-based multisensor cloud phase classifier, Geophysical Research Letters, 34, L22809, https://doi.org/10.1029/2007GL031008, 2007.

Vivekanandan, J., Ghate, V. P., Jensen, J. B., Ellis, S. M., and Schwartz, M. C.: A Technique for Estimating Liquid Droplet Diameter and Liquid Water Content in Stratocumulus Clouds Using Radar and Lidar Measurements, Journal of Atmospheric and Oceanic Technology, 37, 2145–2161, https://doi.org/10.1175/JTECH-D-19-0092.1, 2020.

Wang, Z., and Sassen, K.: Cloud Type and Macrophysical Property Retrieval Using Multiple Remote Sensors, Journal of Applied Meteorology, 40, 1665–1682, https://doi.org/10.1175/1520-0450(2001)040<1665>2.0.CO;2, 2001.

Wood, R.: Drizzle in stratiform boundary layer clouds. Part I: Vertical and horizontal structure, J. Atmos. Sci., 62, 3011–3033, https://doi.org/10.1175/JAS3529.1, 2005.

Wu, P., Dong, X., Xi, B., Tian, J., and Ward, D. M.: Profiles of MBL cloud and drizzle microphysical properties retrieved from ground-based observations and validated by aircraft in situ measurements over the Azores, Journal of Geophysical Research: Atmospheres, 125, e2019JD032205, https://doi.org/10.1029/2019JD032205, 2020.

Xi, B., Dong, X., Zheng, X., and Wu, P.: Cloud phase and macrophysical properties over the Southern Ocean during the MARCUS field campaign, Atmospheric Measurement Techniques, 15, 3761–3777, https://doi.org/10.5194/amt-15-3761-2022, 2022.

Zheng, X., Xi, B., Dong, X., Wu, P., Logan, T., and Wang, Y.: Environmental effects on aerosol–cloud interaction in non-precipitating marine boundary layer (MBL) clouds over the eastern North Atlantic, Atmos. Chem. Phys., 22, 335–354, https://doi.org/10.5194/acp-22-335-2022, 2022.

Zheng, X., Dong, X., Xi, B., Logan, T., and Wang, Y.: Distinctive aerosol–cloud–precipitation interactions in marine boundary layer clouds from the ACE-ENA and SOCRATES aircraft field campaigns, Atmos. Chem. Phys., 24, 10323–10347, https://doi.org/10.5194/acp-24-10323-2024, 2024.